# CRITIQUE-RL: TRAINING LANGUAGE MODELS FOR CRITIQUING THROUGH TWO-STAGE REINFORCEMENT LEARNING

**Zhiheng Xi**[1][*][†]**, Jixuan Huang**[1][*]**, Xin Guo**[1]**, Boyang Hong**[1]**, Dingwen Yang**[1]**, Xiaoran Fan**[1]**,
Shuo Li**[1]**, Zehui Chen**[1]**, Junjie Ye**[1]**, Siyu Yuan**[1]**, Zhengyin Du**[2]**, Jiecao Chen**[2]**, Rui Zheng**[1]**,
Tao Gui**[1][†]**, Qi Zhang**[1]**, Xuanjing Huang**[1]

[1]Fudan University [2]ByteDance
zhxi22@m.fudan.edu.cn, tgui@fudan.edu.cn

## ABSTRACT

Training critiquing language models [1] to assess and provide feedback on model outputs is a promising way to improve LLMs for complex reasoning tasks. However, existing approaches typically rely on stronger supervisors for annotating critique data. To address this, we propose Critique-RL, an online RL approach for developing critiquing language models without stronger labeling. Our approach operates on a two-player paradigm: the actor generates a response, the critic provides feedback, and the actor refines the response accordingly. We first reveal that relying solely on indirect reward signals from the actor's outputs for RL optimization often leads to unsatisfactory critics: while their helpfulness (i.e., providing constructive feedback) improves, the discriminability (i.e., determining whether a response is high-quality or not) remains poor, resulting in marginal performance gains. To overcome this, Critique-RL adopts a two-stage optimization strategy. In stage I, it reinforces the discriminability of the critic with direct rule-based reward signals; in stage II, it introduces indirect rewards based on actor refinement to improve the critic's helpfulness, while maintaining its discriminability via appropriate regularization. Extensive experiments across various tasks and models show that Critique-RL delivers substantial performance improvements. For example, it achieves a $9.02\%$ gain on in-domain tasks and a $5.70\%$ gain on out-of-domain tasks for Qwen2.5-7B, highlighting its potential. [2]

## 1 INTRODUCTION

With the development of large language models (Ouyang et al., 2022; OpenAI, 2023; Touvron et al., 2023; Jiang et al., 2023; Dubey et al., 2024), providing reliable supervision for them has become a critical research challenge (Bowman et al., 2022; Saunders et al., 2022; Chen et al., 2025), especially for tasks that are difficult even for humans, such as complex reasoning, sequential decision-making, and coding (Shinn et al., 2023; Snell et al., 2024; Qu et al., 2024; Kumar et al., 2024; Xi et al., 2024a). This problem is often referred to as scalable oversight (Bowman et al., 2022). One effective method for scalable oversight is to train critiquing language models to assess and provide feedback to

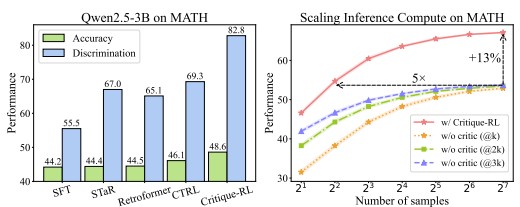

Figure 1: **Left:** Critique-RL achieves better performance and discrimination on MATH. **Right:** Inference compute scaling for Critique-RL, with @2k and @3k indicating sampling amounts that are 2 times and 3 times the x-axis value, respectively. Critique-RL improves the performance ceiling and is more compute-efficient.

---

[*]Equal contribution.

[†]Corresponding author.

[1]It can also be referred to as a critique model or critic.

[2]Our code are available at https://github.com/WooooDyy/Critique-RL

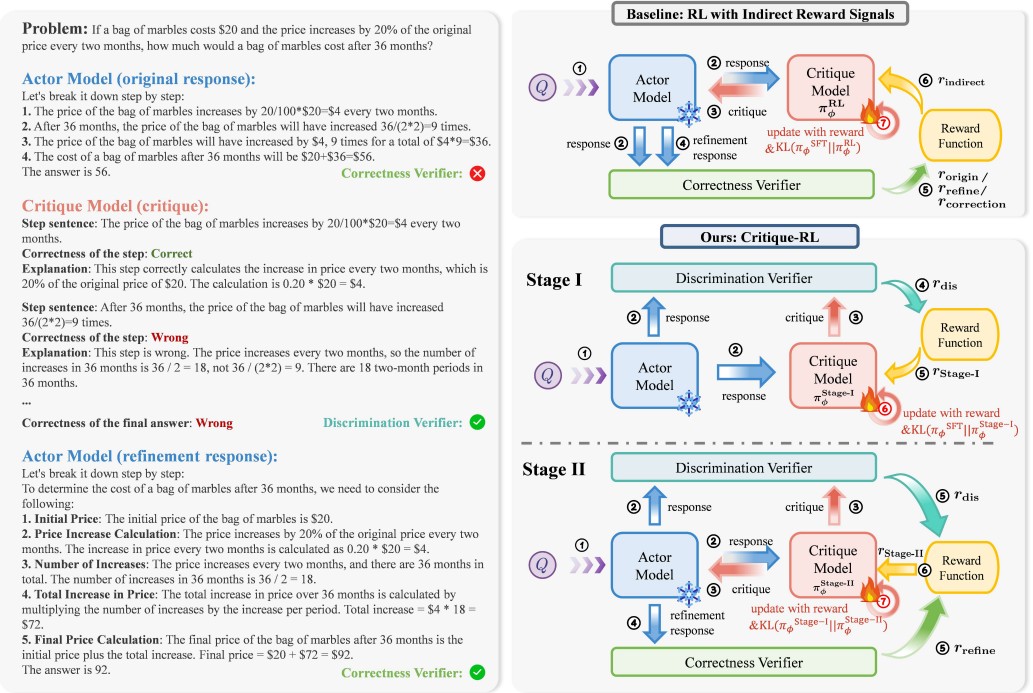

Figure 2: **Left:** A case illustrating the two-player actor-critic interaction, including the original response from the actor, the critique from the critic, and the refinement from the Actor. **Right:** Overview of our method and its comparison with baseline RL. The snowflake icon ❄ on the Actor indicates that it is fixed, while the fire icon 🔥 on the Critic indicates that it will be updated. Our method employs a two-stage RL process. It optimize discriminability of critique models in Stage I, and optimize helpfulness while maintaining discriminability in Stage II.

model outputs (Welleck et al., 2023; Akyürek et al., 2023; Xi et al., 2024b; Yao et al., 2024). Based on this feedback, actor models can refine and optimize their behavior or outputs.

Existing work in training critique models typically assumes a stronger supervisor to provide labeled critique data, which is often expensive and difficult to scale (Saunders et al., 2022; Xi et al., 2024b; Bowman et al., 2022). Moreover, the data labeled by the supervisor often differs significantly from the learner's output distribution (Kumar et al., 2024). Another line of work does not train the model but instead relies on the model's inherent abilities, using prompt engineering to elicit its critiquing abilities (Bai et al., 2022; Madaan et al., 2023; Dhuliawala et al., 2024). However, such methods typically assume an oracle verifier during testing, allowing the critique model to bypass discrimination (i.e., determining whether a response is high-quality) and instead focus only on offering helpful feedback for revision (Xi et al., 2024b; Gou et al., 2024). Without the oracle verifier, they often meet performance bottleneck (Huang et al., 2024).

In this work, we aim to develop critiquing language models without relying on stronger labeling or an oracle reward function during testing. To this end, we propose Critique-RL, an online RL approach based on two-player actor-critic interaction (Yao et al., 2024; Xi et al., 2024b) for developing critique models. In our approach, there are two main roles: the actor and critic. The critic assesses (discriminability) and provides natural language feedback (helpfulness) for the actor's output, and the actor performs refinement accordingly (Saunders et al., 2022).

To build our method, we first use the correctness of the actor's two attempts to shape the reward signals for the RL optimization of critique models (§4.1), following approaches like Retroformer (Yao et al., 2024) and CTRL (Xie et al., 2025), where such indirect signals are shown to reflect the quality of critiques. However, this approach fails to develop satisfactory critique models, i.e., with low performance. Delving into the optimization process, we reveal that while the helpfulness of the

critique models improves, their discriminability is not well optimized, leading to an optimization bottleneck and even a collapse of RL training.

To address the challenges, Critique-RL employs a two-stage RL approach (§4.2). Specifically, as shown in Figure 2, in the first stage, we optimize the discriminability of the critique models using direct rule-based reward signals. In the second stage, we introduce indirect rewards based on the correctness of actor refinement to enhance the helpfulness, while using appropriate regularization to maintain their discriminability. In-depth training dynamics shows that our method addresses the training collapse and stably optimizes both discriminability and helpfulness. Extensive experiments show that our method outperforms baselines across different models and tasks, yielding a $9.02\%$ improvement on in-domain tasks and $5.70\%$ improvement on out-of-domain tasks for Qwen2.5-7B. It is also noteworthy that critique models trained with our method can generalize to unseen tasks, demonstrating its promise for scalable oversight.

In summary, our main contributions are:

1. Delving into the RL optimization process, we reveal that solely depending on indirect reward signals of actor's output correctness cannot develop effective critique models, which poses conflict and optimization challenges between the discriminative and feedback capabilities of critics.

2. We then propose Critique-RL, a novel two-stage RL approach to develop critique models for providing accurate assessment and helpful feedback for model outputs.

3. We perform in-depth experiments, ablation and analysis to show the effectiveness and stability of our method. We hope our work provides insights for the community.

## 2 RELATED WORK

**Prompt engineering for eliciting critiquing ability from language models.** As a key technique for scalable oversight (Bowman et al., 2022), many previous works have explored the use of prompt engineering to elicit the critiquing and reflection abilities of LLMs (Bai et al., 2022; Madaan et al., 2023; Ye et al., 2023; Dhuliawala et al., 2024). These methods typically rely on an oracle verifier including answer matching or external tools at test time for discrimination, allowing the LLM to focus solely on providing natural language feedback (Xi et al., 2024b; Huang et al., 2024). However, in the absence of an external verifier, even SOTA models face significant challenges (Saunders et al., 2022; Welleck et al., 2023; Xu et al., 2024; Huang et al., 2024). In this work, we do not assume an oracle verifier; instead, we train critique models through RL to optimize both discriminability and the ability to provide helpful feedback.

**Fine-tuning language models for critiquing.** Previously, a line of work has explored fine-tuning-based approaches for training critique models (Saunders et al., 2022; Bowman et al., 2022; Xi et al., 2024b). However, these methods primarily rely on a stronger supervisor for data annotation, which is costly and difficult to scale (Xi et al., 2024b). To address this issue, some researchers have proposed self-improvement-based methods to train models for self-critiquing (Tang et al., 2025; Zheng et al., 2024; Yuan et al., 2025b). Unlike these approaches, we adopt a two-player paradigm and train a separated critique model through RL.

**Reinforcement learning for language models.** RL has become an essential component of LLM post-training, such as RLHF for alignment (Ouyang et al., 2022; Zheng et al., 2023; Wang et al., 2024; Shao et al., 2024; Chen et al., 2024). Additionally, various works have leveraged RL to enhance language models' performance in reasoning (Snell et al., 2024; Kumar et al., 2024; Xi et al., 2024a), coding (Kumar et al., 2024), and decision-making tasks (Shinn et al., 2023; Xi et al., 2025b). Furthermore, some studies explore using RL to improve LM's ability for self-reflection and self-correction (McAleese et al., 2024; Kumar et al., 2024; Welleck et al., 2023; Shinn et al., 2023; Xu et al., 2024; Ye et al., 2023; Yuan et al., 2025a). Other methods, such as Retroformer (Yao et al., 2024) and CTRL (Xie et al., 2025), leverage indirect reward signals to optimize critique model's helpfulness, targeting decision-making tasks and coding tasks, respectively. However, their RL phase overlooks the joint optimization of discriminability and helpfulness. Different from them, we propose a two-stage Critique-RL approach to optimize both discriminability and helpfulness, effectively developing critique models.

## 3 PRELIMINARIES

### 3.1 THE TWO-PLAYER INTERACTION FRAMEWORK

The multi-agent framework in this work consists of two main roles (Yao et al., 2024; Xi et al., 2024b): the actor model and the critique model. It operates through a response-critique-refinement process.

Specifically, given a question $x$, the actor model is expected to generate an original response $y = \pi_\theta(x)$, which includes both the reasoning trajectory and the final answer. The correctness verifier then provides an oracle reward $r_{\text{oracle}}(x, y)$ to the actor model. Subsequently, the critique model $\pi_\phi$ takes the question-response pair $(x, y)$ as input and produces critique $c = \pi_\phi(x, y)$, which should include assessment of the response correctness (discriminability) and offer constructive natural language feedback (helpfulness). Based on this critique, the actor model generates a refinement response $y^{'} = \pi_\theta(x, y, c)$, and subsequently receives an oracle reward $r_{\text{oracle}}(x, y^{'})$. Using these rewards, i.e., $r_{\text{oracle}}(x, y)$ and $r_{\text{oracle}}(x, y^{'})$, we can design different reward functions $r_{\text{c}}(\cdot)$ for critique models, which will be shown in §4.

### 3.2 POLICY GRADIENT FOR LLMS

Policy gradient methods (Sutton et al., 1999), e.g., REINFORCE (Ahmadian et al., 2024; Kumar et al., 2024), are common techniques to perform RL on LLMs (Ouyang et al., 2022; Xi et al., 2025a). For the policy critique model $\pi_\phi$ parameterized by $\phi$, the objective of policy gradient is to find an optimal policy that maximizes the reward function $r_{\text{c}}(\cdot)$. It is typically expressed as maximizing:

$$\mathbb{E}_{c\sim\pi_\phi(\cdot|x,y),y'\sim\pi_\theta(x,y,c)}[r_{\text{c}}(x, y, c, y^{'})], \tag{1}$$

where $\mathbb{E}_{c\sim\pi_\phi(\cdot|x,y),y'\sim\pi_\theta(x,y,c)}$ denotes the expectation over the critique sampled from the critic $\pi_\phi$ and the refinement response sampled from the actor $\pi_\theta$. This gradient is used to optimize the critique model via gradient ascent. The positive critique is "reinforced" by increasing its probability.

### 3.3 EVALUATION METRICS

To evaluate the performance of the critique model, we consider the following metrics: (1) **Acc@Refine**: the accuracy of the actor model's refinement response; (2) $\boldsymbol{\Delta}$: the improvement in the actor model's accuracy between the original and refinement response, which measures the effectiveness of the critique model; (3) $\boldsymbol{\Delta^{c\rightarrow i}}$: the change rate from an originally correct response to an incorrect refinement response. A lower value is better; (4) $\boldsymbol{\Delta^{i\rightarrow c}}$: the change rate from an originally incorrect response to a correct refinement response. A higher value is better; (5) **Acc@Dis**: a direct metric to measure the discriminability of the critique model, which quantifies the accuracy of whether the correctness accessed by the critic aligns with the true correctness of the original response.

## 4 METHODOLOGY

### 4.1 MOTIVATING FINDINGS: RL WITH INDIRECT REWARD SIGNALS IS INSUFFICIENT FOR TRAINING SATISFACTORY CRITIQUE MODELS

In the two-player actor-critic framework (Yao et al., 2024; Xi et al., 2024b), a natural and intuitive way to optimize the critiquing language models is to shape the reward signals derived from the actor's two attempts (original and refinement responses). We explore several reward shaping approaches, demonstrate their failure modes, and investigate why they fail to incentivize satisfactory critiquing ability.

**Analysis setups: data, models, and training methods.** Our preliminary experiments are on GSM8K (Cobbe et al., 2021), and the backbone model is Qwen2.5-3B (Team, 2024). Following previous work (Xi et al., 2024b), we train an actor model capable of generating responses and reasonably following critiques. To build the SFT dataset for initializing a base critique model, we prompt Qwen2.5-3B-Instruct to obtain critique data $\mathcal{D}_{\text{SFT}} = \{x, y, c\}_{i=1}^{|\mathcal{D}_{\text{SFT}}|}$, rather than using annotations from SOTA commercial models like GPT-4o (OpenAI, 2023). We filter the critique data based on the correctness of refinement to ensure the quality.

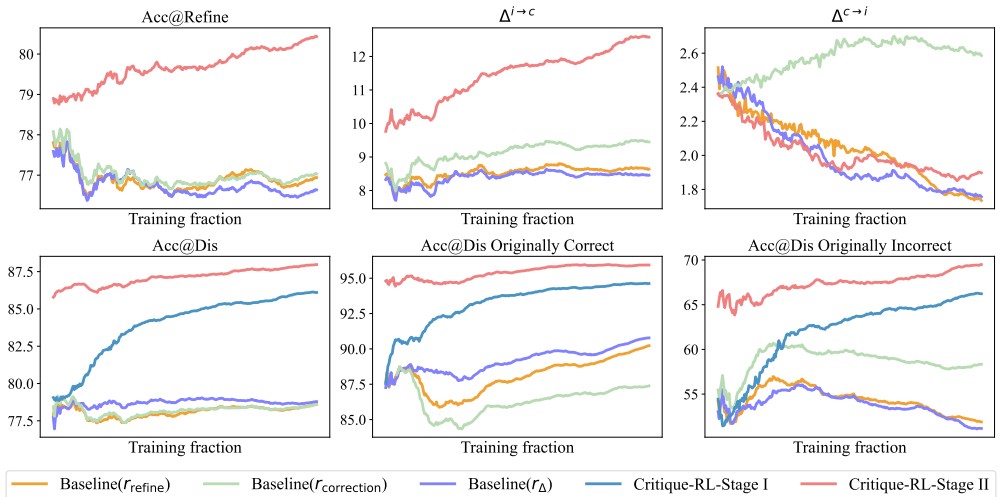

Figure 3: Training dynamics of preliminary experiments. "Acc@Dis Originally Correct" and "Acc@Dis Originally Incorrect" refer to the discrimination accuracy of originally correct and incorrect responses, respectively. Baselines using indirect reward signals to optimize helpfulness tend to exhibit overly conservative or aggressive behavior as the discriminability is not well optimized. In contrast, our Critique-RL optimizes discriminability in Stage I, and optimizes helpfulness while maintaining discriminability in Stage II, achieving better in Acc@Refine, $\mathbf{\Delta}^{c \to i}$ and $\mathbf{\Delta}^{i \to c}$.

Next, we train the critique model $\pi_\phi$ using the SFT loss:

$$\mathcal{L}_{\text{SFT}}(\phi) = \mathbb{E}_{(x,y,c) \sim \mathcal{D}_{\text{SFT}}}\Big[ \log \pi_\phi(c|x,y) \Big]. \tag{2}$$

We then employ policy gradient (Sutton et al., 1999) to maximize:

$$\mathbb{E}_{c \sim \pi_\phi^{\text{RL}}(\cdot|x,y), y' \sim \pi_\theta(\cdot|x,y,c)}\Big[ r_c(x,y,c,y') - \beta \text{KL}(\pi_\phi^{\text{SFT}}(c|x,y)||\pi_\phi^{\text{RL}}(c|x,y)) \Big], \tag{3}$$

where $\pi_\theta$ is the fixed actor model, $\pi_\phi^{\text{SFT}}$ is the SFT model. Each $x$ is a query sampled from the RL dataset $\mathcal{D}_{\text{RL}}$, $y$ is the original response. $\text{KL}(\cdot||\cdot)$ means the KL-divergence which constrains the distance between the RL model and the SFT model, and $\beta$ is a scaling factor. $r_c(\cdot)$ is the reward function for critique models. Here, with $r_{\text{oracle}}$ being the oracle reward function that verifies the correctness of an actor response, $r_c(\cdot)$ can be $r_{\text{refine}}$ which represents the correctness of the refinement:

$$r_{\text{refine}}(x,y,c,y') = r_{\text{oracle}}(x,y'), \tag{4}$$

or it can be $r_{\mathbf{\Delta}}$ which represents the difference in correctness between the actor's two attempts:

$$r_{\mathbf{\Delta}}(x,y,c,y') = r_{\text{oracle}}(x,y') - r_{\text{oracle}}(x,y). \tag{5}$$

Moreover, we also include $r_{\text{correction}}$ as $r_c(\cdot)$ for reinforcing the ability to correct incorrect responses:

$$r_{\text{correction}}(x,y,c,y') = \begin{cases} 1.0, r_{\text{oracle}}(x,y) = 0 \text{ and } r_{\text{oracle}}(x,y') = 1, \\ 0.2, r_{\text{oracle}}(x,y) = 1 \text{ and } r_{\text{oracle}}(x,y') = 1, \\ 0.0, r_{\text{oracle}}(x,y') = 0. \end{cases} \tag{6}$$

**Empirical findings and behavior analysis.** We illustrate the training dynamics during RL in Figure 3. Optimizing with $r_{\text{refine}}$ and $r_{\mathbf{\Delta}}$ can reduce $\mathbf{\Delta}^{c \to i}$, preventing originally correct responses from being altered incorrectly, but its $\mathbf{\Delta}^{i \to c}$ is not significantly optimized, meaning its error correction performance is not good enough. This phenomenon reveals that the critique model is overly **conservative**, encouraging the actor to not change its answers. As a result, the final Acc@Refine is not satisfactory.

In contrast, optimizing with $r_{\text{correction}}$ improves $\mathbf{\Delta}^{i \to c}$, but fails to effectively reduce $\mathbf{\Delta}^{c \to i}$. This means it often provides more **aggressive** suggestions, encouraging the actor model to correct incorrect responses, but it also introduces a greater risk of turning originally correct answers into incorrect ones. Similarly, the final Acc@Refine is also not satisfactory.

---

**Algorithm 1:** Critique-RL

---

**Input:** Actor model $\pi_\theta$, base critique model $\pi_\phi$, SFT dataset $\mathcal{D}_{\text{SFT}}$, RL dataset $\mathcal{D}_{\text{RL}}$, function that extracts the correctness of a response judged by a critique $f$, oracle reward function $r_{\text{oracle}}$, discrimination reward function $r_{\text{dis}}$.

**Procedure** Supervised Fine-tuning:

$\quad \pi_\phi^{\text{SFT}} \leftarrow \pi_\phi$;

$\quad$ Update $\pi_\phi^{\text{SFT}}$ by minimizing $\mathcal{L}_{\text{SFT}}(\phi) = \mathbb{E}_{(x,y,c) \sim \mathcal{D}_{\text{SFT}}}\Big[\log \pi_\phi(c|x,y)\Big]$;

**Procedure** Critique-RL Stage I: optimizing discriminability through direct reward signals.

$\quad \pi_\phi^{\text{Stage-I}} \leftarrow \pi_\phi^{\text{SFT}}$;

$\quad$ **for** batch in $\mathcal{D}_{RL}$ **do**

$\quad\quad$ **for** $x$ in batch **do**

$\quad\quad\quad$ Generate $y$ and $c$ with $\pi_\theta$ and $\pi_\phi^{\text{Stage-I}}$;

$\quad\quad\quad$ Compute discrimination reward with $r_{\text{dis}}(x,y,c) = \mathbb{1}\Big(f(x,y,c) = r_{\text{oracle}}(x,y)\Big)$;

$\quad\quad$ **end**

$\quad\quad$ Update $\pi_\phi^{\text{Stage-I}}$ by maximizing

$\quad\quad\quad \mathbb{E}_{c \sim \pi_\phi^{\text{Stage-I}}(\cdot|x,y)}\Big[r_{\text{dis}}(x,y,c) - \beta \text{KL}(\pi_\phi^{\text{SFT}}(c|x,y)||\pi_\phi^{\text{Stage-I}}(c|x,y))\Big]$;

$\quad$ **end**

**Procedure** Critique-RL Stage II: optimization helpfulness while maintaining discriminability.

$\quad \pi_\phi^{\text{Stage-II}} \leftarrow \pi_\phi^{\text{Stage-I}}$;

$\quad$ **for** batch in $\mathcal{D}_{RL}$ **do**

$\quad\quad$ **for** $x$ in batch **do**

$\quad\quad\quad$ Generate $y$, $c$ and $y^{'}$ with $\pi_\theta$ and $\pi_\phi^{\text{Stage-II}}$;

$\quad\quad\quad$ Compute discrimination reward with $r_{\text{dis}}(x,y,c) = \mathbb{1}\Big(f(x,y,c) = r_{\text{oracle}}(x,y)\Big)$;

$\quad\quad\quad$ Compute refinement reward with $r_{\text{refine}} = r_{\text{oracle}}(x,y^{'})$;

$\quad\quad$ **end**

$\quad\quad$ Update $\pi_\phi^{\text{Stage-II}}$ by maximizing $\mathbb{E}_{c \sim \pi_\phi^{\text{Stage-II}}(\cdot|x,y),y' \sim \pi_\theta(\cdot|x,y,c)}\Big[r_{\text{refine}} + \beta_1 r_{\text{dis}}(x,y,c) -$

$\quad\quad \beta_2 \text{KL}(\pi_\phi^{\text{Stage-I}}(c|x,y)||\pi_\phi^{\text{Stage-II}}(c|x,y))\Big]$.

$\quad$ **end**

---

**Analyzing underlying reasons for the failure modes.** To reveal the reasons behind the above failure modes, we also visualize the discrimination performance of the critiquing language models during RL in Figure 3. We find that as RL progresses, all three reward functions $r_{\text{refine}}$, $r_{\mathbf{\Delta}}$ and $r_{\text{correction}}$ fail to optimize discriminability effectively. For originally correct and incorrect responses, they can only optimize the judgment for one, while the ability to judge the other is reduced. This may be because both of the indirect reward functions are based on the actor's responses, targeting helpfulness and overlooking discriminability. This motivates the proposal of our method.

## 4.2 Two-Stage Critique-RL

**Key challenges.** Based on the previous analysis, we have identified two key challenges in RL for critiquing language models: (1) optimizing the discriminability of critique models to improve their accuracy in judging both correct and incorrect original responses; (2) improving the quality of the model's feedback, i.e., helpfulness, while maintaining its discriminability, to prevent the issues of being overly aggressive or overly conservative.

**Method overview.** To address the above challenges, we propose the two-stage Critique-RL. In the first stage, our method explicitly optimizes the discriminability of the critique model using direct reward signals. We then use the resulting model $\pi_\phi^{\text{Stage-I}}$ as the initialization for the second stage. In the second stage, we introduce a reward function based on the actor's response to optimize the critic's

helpfulness, while also incorporating appropriate regularization to maintain its discriminability. We illustrate our method in Figure 2 and the algorithm is summarized in Algorithm 1.

**Stage I: optimizing discriminability through direct reward signals.** We decouple the discriminability and helpfulness of the critique models (Saunders et al., 2022; Chen et al., 2024). In Stage I, we shape the reward based solely on the actor's original response. Given $(x, y)$, critique models are prompted to give correctness judgments for each step, and also provide a judgment for the final answer. Based on this, we define the discriminability reward function of the critique models as:

$$r_{\text{dis}}(x, y, c) = \mathbb{1}\Big(f(x, y, c) = r_{\text{oracle}}(x, y)\Big), \tag{7}$$

where $f(x, y, c)$ is the critique model's judgment of the correctness of the original response. $\mathbb{1}(\cdot)$ is indicator function that returns 1 only when the condition inside the parentheses holds, and 0 otherwise. Based on this, our Stage I RL maximizes:

$$\mathbb{E}_{c \sim \pi_\phi^{\text{Stage-I}}(\cdot|x,y)}\Big[r_{\text{dis}}(x, y, c) - \beta\text{KL}(\pi_\phi^{\text{SFT}}(c|x,y)||\pi_\phi^{\text{Stage-I}}(c|x,y))\Big], \tag{8}$$

where the KL divergence with the SFT model is still used to stabilize the training. As shown in Figure 3, our Stage I RL can effectively and stably optimize discriminability, regardless of the correctness of the original response.

**Stage II: optimizing helpfulness while maintaining discriminability.** The goal of the second stage of Critique-RL is to optimize the helpfulness of the critique models without sacrificing their discriminability, thereby avoiding overly conservative or overly aggressive behavior patterns. To achieve this, we introduce a reward function $r_{\text{refine}}$ based on actor refinement correctness. Meanwhile, to preserve the model's discriminability, we retain $r_{\text{dis}}$ and introduce a regularization term based on the KL divergence with the Stage I model $\pi_\phi^{\text{Stage-I}}$. Specifically, we maximize the following objective:

$$\mathbb{E}_{c \sim \pi_\phi^{\text{Stage-II}}(\cdot|x,y), y' \sim \pi_\theta(\cdot|x,y,c)}\Big[r_{\text{refine}} + \beta_1 r_{\text{dis}}(x, y, c) - \beta_2\text{KL}(\pi_\phi^{\text{Stage-I}}(c|x,y)||\pi_\phi^{\text{Stage-II}}(c|x,y))\Big], \tag{9}$$

where $\beta_1$ and $\beta_2$ are scaling factors. As shown in Figure 3, our Stage II effectively optimizes the model's helpfulness, increasing $\mathbf{\Delta}^{i \to c}$ and decreasing $\mathbf{\Delta}^{c \to i}$, ultimately leading to a stable improvement in Acc@Refine and $\mathbf{\Delta}$. Our method also performs strongly on the test set (see §5).

## 5 EXPERIMENTS

### 5.1 EXPERIMENTAL SETUP

**Datasets.** Focusing on mathematical reasoning tasks, we select 5 different commonly-used tasks, including free-from and multiple-choice. Following Ding et al. (2025), we construct training set with the train-split of MATH (Hendrycks et al., 2021), GSM8K (Cobbe et al., 2021), AQUA (Ling et al., 2017). The testset of the three tasks are used as in-domain testset, while the test-split of SVAMP (Patel et al., 2021), TheoremQA (Chen et al., 2023), are used as our OOD (out-of-domain) testset.

**Models and baselines.** Our experiments are mainly conducted on Qwen2.5 series (Team, 2024), i.e., Qwen2.5-3B and Qwen2.5-7B. Besides, we also conduct experiments on other models like Qwen2.5-72B, Llama3.2 (Dubey et al., 2024) and DeepSeek-R1-Distill-Qwen-7B (DeepSeek-AI, 2025) (see Appendix C and Section 6). We include several baselines: (1) SFT which fine-tunes models with critique data. (2) STaR (Zelikman et al., 2022) which iteratively fine-tunes critique models on self-generated data and filtered based on the refinement correctness of the actor. (3) RL baselines that leverages indirect outcome-based reward as baselines, i.e., Retroformer (Yao et al., 2024) which uses PPO and CTRL (Xie et al., 2025) which uses GRPO.

**Implementation details.** All experiments are conducted on 8 NVIDIA A800 GPUs. To initialize an actor that can reason and refine based on the critiquing feedback, we follow Ding et al. (2025); Xi et al. (2024b) to construct a dataset of $21,973$ reasoning traces and $12,000$ refinement responses. For critique data, we construct a set of $6,000$ examples, with $2,000$ examples in each training task. For fine-tuning actors, we set epoch to 3 and learning rate to $5e - 6$, and remains fixed during further training phase; for fine-tuning critics, we set epoch to 5 and learning rate to $5e - 6$. We use the same base model for the actor and the critique model. For STaR and RL, we perform SFT to obtain an initialized model. In RL, we set KL coefficient to $0.01$. In Critique-RL, we use RLOO as our base

Table 1: Main results. The best performance is in **bold** and underlined, while the second-best performance is underlined. Our method is marked in blue . No Critic means the actor model perform reasoning only, and we report the reasoning performance. For other methods, we report the Acc@Refine performance for the acc column.

| Model | Method | MATH | | | GSM8K | | | AQuA | | |
|---|---|---|---|---|---|---|---|---|---|---|
| | | Acc | Δ | Acc@Dis | Acc | Δ | Acc@Dis | Acc | Δ | Acc@Dis |
| Qwen2.5-3B | No Critic | 36.90 | – | – | 66.03 | – | – | 50.00 | – | – |
| | SFT | 44.24 | 7.34 | 66.51 | 69.14 | 3.11 | 76.34 | 46.46 | −3.54 | 61.97 |
| | STaR | 44.38 | 7.48 | 66.97 | 71.95 | 5.91 | 74.79 | 50.39 | 0.39 | 66.13 |
| | Retroformer | 44.54 | 7.64 | 65.11 | 70.51 | 4.47 | 77.59 | 51.18 | 1.18 | 58.44 |
| | CTRL | 46.14 | 9.24 | 69.29 | 70.58 | 4.55 | 76.71 | 53.54 | 3.54 | 62.20 |
| | Critique-RL | **48.60** | **11.70** | **82.80** | **75.89** | **9.86** | **87.44** | **56.69** | **6.69** | **69.92** |
| Qwen2.5-7B | No Critic | 45.74 | – | – | 75.66 | – | – | 63.39 | – | – |
| | SFT | 51.84 | 6.10 | 67.59 | 78.77 | 3.11 | 79.42 | 59.45 | −3.94 | 68.67 |
| | STaR | 54.06 | 8.32 | 69.71 | 80.52 | 4.85 | 81.03 | 57.87 | −5.51 | 72.18 |
| | Retroformer | 52.34 | 6.60 | 68.03 | 80.82 | 5.16 | 77.05 | 63.39 | 0.00 | 70.56 |
| | CTRL | 53.86 | 8.12 | 71.42 | 81.35 | 5.69 | 83.44 | 64.96 | 1.57 | 71.66 |
| | Critique-RL | **58.40** | **12.66** | **85.20** | **87.72** | **12.05** | **90.43** | **65.75** | **2.36** | **78.09** |

algorithm as it performs well and does not require a value model. In Stage II, $\beta_1$ is set to $0.2$. We train the critique model for $500$ steps at each stage and report best results. During evaluation, the temperature is set to $0$. For inference-compute scaling and Pass@$K$, we set temperature to $0.7$.

## 5.2 MAIN RESULTS

**Generally, critique models can significantly improve actor's reasoning performance.** The results in Table 1 demonstrate that when introducing critique models, the actor's reasoning performance can be boosted by a large margin. For example, in the MATH task, even the SFT Baseline outperforms the model without a critic by $7.34$ and $6.10$ points on the 3B and 7B models, respectively. This suggests that critique models are an effective scalable oversight method, as discussed in Saunders et al. (2022); McAleese et al. (2024).

**RL-based methods outperforms fine-tuning-based ones.** Both SFT and STaR methods lead to promising critique models, but in most cases, online RL-based methods perform better, especially our Critique-RL. For instance, on the 3B model, our method surpasses the SFT method by an average of $7.11$ points on accuracy across three datasets. It is worth noting that on AQuA, fine-tuning-based SFT and STaR may lead to negative impact on performance, while our method provides significant positive improvements. This reveals that online RL methods have greater potential and adaptability in eliciting the model's critiquing ability, similar to the findings in McAleese et al. (2024).

**Critique-RL consistently outperforms other baselines in discrimination and final accuracy.** In terms of discrimination, our method also significantly outperforms other baselines, such as surpassing CTRL by $5.31$, $6.36$ points for 3B and 7B models on GSM8K, respectively. This reveals that our discrimination-related reward shaping can effectively optimizes discriminability. Thanks to this and the helpfulness reward design in the second stage, our method shows a significant improvement in final performance compared to other baselines. For example, on the 7B model, our method outperforms Retroformer by an average of $5.11$ and $12.69$ points on accuracy and discriminability, across three datasets.

## 5.3 ITERATIVE IMPROVEMENT OF CRITIQUE-RL

Furthermore, we validate the iterative improvement capability of Critique-RL through two key aspects: (1) Iterative refinement process: During the $i$-th iteration, the critic generates critique $c_i = \pi_\phi(x, y_0, c_1, ..., c_{i-1}, y_{i-1})$, while the actor produces the refined response $y_i = \pi_\theta(x, y_0, c_1, ..., y_{i-1}, c_i)$ accordingly. (2) Iterative training process: We alternately conduct the two-stage training of Critique-RL (Stage I and

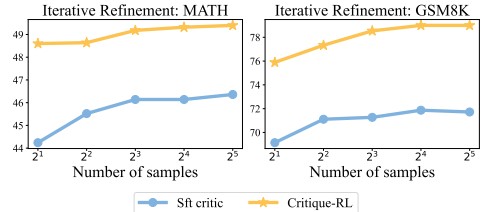

Figure 4: Results of iterative critique-refinement of Critique-RL using Qwen2.5-3B.

Stage II) to optimize the critique model. The detailed results are shown in Figure 4 and Table 2, respectively.

First, as demonstrated in Figure 4, through iterative critique and refinement, the model exhibits consistent Acc gains on Qwen2.5-3B, with each iteration achieving measurable improvements. Second, iterative training leads to further performance enhancement, with detailed results using Qwen2.5-3B on MATH dataset shown in Table 2. Specifically, both Stage I and Stage II of Critique-RL demonstrate consistent improve-

Table 2: Results of iterative training of Critique-RL using Qwen2.5-3B on MATH.

| Method | | Acc | $\Delta$ | Acc@Dis |
|---|---|---|---|---|
| | No Critic | 36.9 | – | – |
| | SFT | 44.2 | 7.3 | 66.5 |
| Critique-RL | Iteration 1, Stage I | 45.9 | 9.0 | 78.7 |
| | Iteration 1, Stage II | 48.6 | 11.7 | 82.8 |
| | Iteration 2, Stage I | 49.5 | 12.6 | 85.0 |
| | Iteration 2, Stage II | **51.0** | **14.1** | **86.5** |

ment in Acc and Acc@Dis metrics. Compared to the first iteration, the second iteration improves by 2.40 and 3.68 points on accuracy and discriminability.

# 6 DISCUSSION AND ANALYSIS

**Ablation on different stages.** We conduct ablation experiments to validate the importance of different components. The results are shown in Table 3. Both Stage I and Stage II are crucial, and removing either of them leads to a performance drop. This indicates that optimizing both discriminability and helpfulness is essential in developing critique models.

**Ablation on reward design for Stage II.** Next, we perform a deeper analysis of the reward design in Stage II. First, if we remove the discrimination-related $r_{\text{dis}}$ and KL-based regularization

Table 3: Ablation study using Qwen2.5-3B. We report the Acc@Refine. "w/o" means without; "Stage II w/o discrimination" means in Stage II, we remove $r_{\text{dis}}$ and $\text{KL}(\pi_\phi^{\text{Stage-I}}||\pi_\phi^{\text{Stage-II}})$; "Stage II w/ $r_\Delta$" and "Stage II w/ $r_{\text{correction}}$" mean replacing the $r_{\text{refine}}$ with the corresponding reward function.

| Method | MATH | | AQuA | |
|---|---|---|---|---|
| | Acc@Refine | Acc@Dis | Acc@Refine | Acc@Dis |
| Critique-RL (Ours) | **48.6** | **82.8** | **56.7** | **69.9** |
| -w/o Stage I | 47.6 | 79.7 | 53.9 | 66.5 |
| -w/o Stage II | 45.9 | 78.7 | 54.7 | 68.2 |
| -Stage II w/o discrimination | 47.3 | 77.7 | 53.5 | 61.6 |
| -Stage II w/ $r_\Delta$ | 48.2 | 82.6 | 53.9 | 68.4 |
| -Stage II w/ $r_{\text{correction}}$ | 47.7 | 82.0 | 54.7 | 68.4 |

$\text{KL}(\pi_\phi^{\text{Stage-I}}||\pi_\phi^{\text{Stage-II}})$, the discriminability and accuracy suffer a significant drop. This further emphasizes that when optimizing for helpfulness, it is crucial to maintain the model's discrimination ability. Second, when we replace the reward function $r_{\text{refine}}$ in Stage II with another reward function, i.e., $r_\Delta$ and $r_{\text{correction}}$, we observe a slight performance drop. This may be because $r_{\text{refine}}$ directly optimizes the Acc@Refine metric, which aligns most closely with the test-time scenario.

**Analysis of helpfulness when the oracle verifier is available.** Many previous works have relied on an external oracle verifier to assess the actor's reasoning results (Bai et al., 2022; Madaan et al., 2023; Ye et al., 2023; Dhuliawala et al., 2024). In this scenario, the model's judgment ability is isolated, allowing us to better evaluate the critique model's helpfulness. We conduct relevant experiments, and the results are shown in Figure 5. We find that when the oracle verifier is available, all baselines show performance improvements. In this case, our method still outperforms others across different datasets and models, indicating that our approach significantly enhances the model's helpfulness. Furthermore, comparisons with other RL baselines reveal that the optimization of discriminability in our method also implicitly contributes to the improvement of helpfulness, suggesting that the two abilities are not entirely independent. This further emphasizes the importance of optimizing both abilities jointly in developing critique models.

**Evaluation of test-time inference compute scaling for Critique-RL.** We investigate whether Critique-RL can be combined with inference-time compute scaling strategy. Following Qu et al. (2024); Snell et al. (2024); Xi et al. (2024b), we leverage the commonly used majority vote (MV@$K$) (Wang et al., 2023) which evaluates whether the most frequent answer among $K$ samples is correct. The results of MATH are shown in Figure 1 and the results of GSM8K are shown in Figure 6 of Appendix F. Compared to the baseline, Critique-RL significantly increases the performance ceiling and shows a more sustained upward trend as inference compute scales. More importantly, performing $K\times$ response-critique-refinement sampling is more compute-efficient than conducting $3K\times$ parallel sampling responses, suggesting the compute-efficiency of Critique-RL.

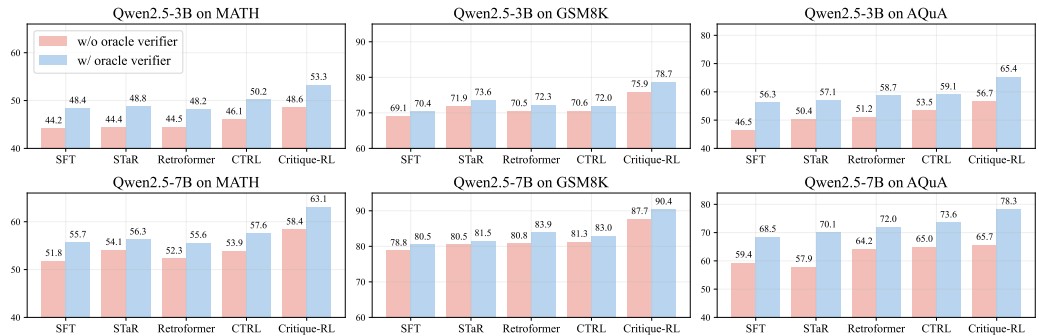

Figure 5: Performance with and without the oracle verifier. When the oracle verifier is available, the model no longer needs to make discrimination and just needs to provides useful feedback. This allows us to evaluate the model's helpfulness more accurately.

**Generalization to OOD tasks.** We also validate the generalization of the models trained by Critique-RL on OOD tasks. The results in Table 4 show that the models trained still delivers significant performance improvements, further demonstrating the potential of this scalable oversight approach.

**More experiments and qualitative analysis.** We conduct extensive experiments to show the effectiveness and working mechanism of Critique-RL, with the detailed results presented in the Appendix: (1) In addition to the Qwen2.5 series (Team, 2024), we evaluate our method on different architectures including Llama3.2 (see Appendix C). (2) We compare Critique-RL with other refinement methods including Self-Refine (Madaan et al., 2023), SuperCorrect (Yang et al., 2024) and Critic-Cot (Zheng et al., 2024), and the results are presented in Appendix E. (3) We also perform test-time scaling analysis of sampling multipe refinement on the same response, with results presented in Appendix F. (4) We conduct experiments on summarization tasks using CNN/DailyMail (Hermann et al., 2015) dataset to investigate our method's generalization ability on open-ended tasks where rule-based verifier cannot be directly applied, the results are in Appendix G. (5) We perform a qualitative analysis on how Critique-RL works and provide several examples in Appendix J.

Table 4: Out-of-domain evaluation of Critique-RL.

| Model | Method | SVAMP | | TheoremQA | |
|---|---|---|---|---|---|
| | | Acc | Pass@10 | Acc | Pass@10 |
| Qwen2.5-3B | No Critic | 70.7 | 92.0 | 15.1 | 34.8 |
| | SFT | 74.7 | 95.7 | 15.3 | 36.1 |
| | Retroformer | 75.0 | 96.0 | 16.1 | 37.0 |
| | CTRL | 76.0 | 95.7 | 15.8 | 36.5 |
| | Critique-RL | **78.3** | **96.3** | **16.8** | **37**.8 |
| Qwen2.5-7B | No Critic | 80.3 | 95.7 | 19.4 | 39.8 |
| | SFT | 83.0 | 95.7 | 20.5 | 41.9 |
| | Retroformer | 84.0 | 96.0 | 20.0 | 42.3 |
| | CTRL | 85.1 | 96.7 | 21.1 | 42.9 |
| | Critique-RL | **89.7** | **97.0** | **21.4** | **43.0** |

# 7 CONCLUSION

In this paper, we propose Critique-RL, an RL approach for developing critique models. Through in-depth analysis, we highlight the importance of explicitly optimizing model discriminability and propose a two-stage RL approach that effectively optimizes both discriminability and helpfulness. We validate its stability and superiority through detailed experiments, and further uncover its working mechanism through ablation studies and analyses. We hope that our work can provide insights for the scalable oversight community of language models.

## ETHICS STATEMENT

This paper presents Critique-RL, a novel two-stage RL approach to develop critiquing language models for providing accurate assessment and helpful feedback for model outputs. We firmly state that this work is intended for ethical and constructive purpose. While no immediate societal harms are evident, this approach enables scalable supervision by training models with minimal direct human oversight. Nevertheless, its potential susceptibility to misuse warrants proactive measures to ensure responsible governance.

## REPRODUCIBILITY STATEMENT

We claim our detailed experiment setting in §5.1. In addition, we upload anonymized versions of our data and code in a Zip file with a Readme file to ensure easy reproduction of all reported results.

## ACKNOWLEDGMENTS

The authors wish to thank the anonymous reviewers for their helpful comments. This work was partially funded by Henan Province Major Industrial "Challenge-Based Innovation" (No. 251000210300), National Natural Science Foundation of China (No.62476061, 62376061, 62576106), Shanghai Rising-Star Program (23QA1400200), and Natural Science Foundation of Shanghai (23ZR1403500).

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

## A  THE USE OF LARGE LANGUAGE MODELS

LLMs are utilized in this manuscript for partial grammatical checks and language polishing. The authors are fully responsible for the final content.

## B  PERFORMANCE ON VARYING ACTOR MODELS WITH DIFFERENT CAPABILITY LEVELS

To further investigate the Critique-RL in varying base models, we conduct two types of experiments. In the first setting, we use a strong reasoning model DeepSeek-R1-Distill-Qwen-7B (DeepSeek-AI, 2025) as our actor model while using Qwen2.5-7B as our critic model. This evaluation setting investigates the generalization of Critique-RL to reasoning models. The results in Table 5 reveal that, besides non-reasoning models (Qwen2.5-3B, Qwen2.5-7B) with structured CoT, our method is also effective for reasoning models with complex CoT structures on both in-domain and out-of-domain tasks, particularly in terms of the Acc@Dis achieved by the critique models. While DeepSeek-R1-Distill-Qwen-7B already performs strongly on MATH-500, critique models can still offer marginal gains in reasoning accuracy. More impressively, on the TheoremQA dataset which spans diverse domains including Math, EECS, Physics and Finance, critique models substantially boost performance, highlighting the strong generalization ability of our approach. Notably, Critique-RL outperforms SFT, Retroformer, and CTRL by 26.75, 28.75, 29.88 points in Acc@Dis, respectively, on the TheoremQA dataset—doubling the performance of these baselines.

Table 5: Performance on DeepSeek-R1-Distill-Qwen-7B as actor.

| Method | In-Domain: MATH-500 | | | OOD: TheoremQA | | |
|---|---|---|---|---|---|---|
| | Acc | Δ | Acc@Dis | Acc | Δ | Acc@Dis |
| No Critic | 84.60 | - | - | 21.63 | - | - |
| SFT | 85.60 | 1.00 | 83.40 | 29.75 | 8.13 | 24.38 |
| Retroformer | 85.80 | 1.20 | 84.80 | 29.38 | 7.75 | 22.38 |
| CTRL | 85.80 | 1.20 | 84.80 | 29.00 | 7.38 | 21.25 |
| Critique-RL | **86.60** | **2.00** | **93.00** | **30.38** | **8.75** | **51.13** |

Table 6: Performance on Qwen2.5-72B-Instruct as actor.

| Method | In-Domain: MATH-500 | | | OOD: TheoremQA | | |
|---|---|---|---|---|---|---|
| | Acc | Δ | Acc@Dis | Acc | Δ | Acc@Dis |
| No Critic | 79.10 | - | - | 21.38 | - | - |
| SFT | 79.20 | 0.10 | 80.20 | 21.63 | 0.25 | 23.00 |
| Retroformer | 79.20 | 0.10 | 80.60 | 21.75 | 0.38 | 21.38 |
| CTRL | 79.40 | 0.30 | 79.40 | 21.50 | 0.13 | 21.13 |
| Critique-RL | **80.30** | **1.20** | **89.20** | **23.50** | **2.10** | **46.63** |

In the second setting, we use Qwen2.5-72B-Instruct as the actor model and Qwen2.5-7B as the critique model to investigate weak-to-strong generalization.

The results in Table 6 show that Critique-RL improves actor performance even in large-scale settings, though with less pronounced gains compared to smaller-actor settings. Nonetheless, it still outperforms baselines on both in-domain and out-of-domain tasks. Notably, our method achieves significantly higher discrimination, confirming the effectiveness of our discrimination-based reward shaping.

## C  PERFORMANCE ON VARYING MODEL SERIES

Table 7: Performance on Llama3.2-3B with GSM8K.

| Method | GSM8K | | |
|---|---|---|---|
| | Acc | Δ | Acc@Dis |
| No Critic | 49.28 | - | - |
| SFT | 50.80 | 1.52 | 68.11 |
| Retroformer | 52.08 | 2.81 | 63.85 |
| CTRL | 52.24 | 2.96 | 66.01 |
| Critique-RL | **52.99** | **3.72** | **75.04** |

To evaluate the effectiveness and generalization capability of Critique-RL, we conduct experiments using the Llama3.2-3B (Dubey et al., 2024) model on the GSM8K dataset. As shown in Table 7, Critique-RL proves effective not only on Qwen2.5 models but also on Llama3.2 models, particularly

in enhancing the discriminability of the critique models. These results highlight the adaptability and robust performance of Critique-RL across different model architectures.

## D PERFORMANCE ON MORE CHALLENGING BENCHMARKS

To further validate the effectiveness of Critique-RL, we conduct experiments using Qwen2.5-7B-Instruct and evaluated on the AIME2024, AIME2025 (AIME, 2025), and GPQA-Diamond (Rein et al., 2024) benchmarks. We used General-Reasoner-7B (Ma et al., 2025) as the actor and constructed a training set of 30k examples based on the WebInstruct-Verified (Ma et al., 2025) dataset for RL training. The experimental results are in Table 8. The experimental results demonstrate that our method significantly improves the critique model's discriminability, with particularly notable improvements on the challenging reasoning datasets AIME2024 and AIME2025. Without fine-grained hyperparameter adjustments, our method outperforms the baseline across all three benchmarks, proving the effectiveness of Critique-RL in complex reasoning scenarios.

Table 8: Performance on challenging benchmarks using Qwen2.5-7B-Instruct as critic.

| Methods | GPQA-Diamond | | | AIME2024 | | | AIME2025 | | |
| --- | --- | --- | --- | --- | --- | --- | --- | --- | --- |
| | Acc | Δ | Acc@Dis | Acc | Delta | Acc@Dis | Acc | Δ | Acc@Dis |
| No Critic | 34.30 | - | - | 11.98 | - | - | 6.67 | - | - |
| SFT | 35.86 | 1.56 | 41.20 | 12.30 | 0.32 | 22.71 | 7.50 | 0.83 | 6.67 |
| Critique-RL | **37.37** | **3.07** | **51.52** | **13.75** | **1.77** | **53.44** | **8.50** | **1.83** | **30.10** |

## E COMPARISON WITH OTHER IMPORTANT REFINEMENT METHODS

To further validate the advantages of Critique-RL over other refinement methods, we conduct evaluations of other refinement methods including Self-Refine (Madaan et al., 2023), SuperCorrect (Yang et al., 2024) and Critic-Cot (Zheng et al., 2024) with Qwen2.5-3B on GSM8K. For a fairer comparison, we train the models in Self-Refine and Critic-CoT using the same dataset(sampled from Qwen2.5-3B-Instruct) as Critique-RL. In terms of SuperCorrect, we choose Deepseek-R1 (DeepSeek-AI, 2025) as the teacher model to create both the Hierarchical Thought Templates and positive critique datasets. The results are presented in Table 9. Critique-RL significantly outperforms all other methods in both Acc and Acc@Dis, surpassing Critic-CoT and SuperCorrect by 5.31 and 3.11 points in terms of Acc, respectively. Moreover, Critique-RL outperforms Self-Refine across refinement iterations, demonstrating its greater effectiveness. Notably, SuperCorrect exhibited poor discriminability, likely because it simply used teacher model data as positive examples and student model data as negative ones for DPO training. Given the GSM8K dataset's simplicity, the student model's output is not consistently inferior to teacher model's, leading to potential impairment to the model's discriminability.

These refinement methods are implemented using SFT (Self-Refine), self-improve (Critic-CoT) or intricate SFT+DPO (SuperCorrect) approaches, wheras Critique-RL employs an online RL methodology, which accounts for its observed performance advantages.

## F MORE TEST-TIME SCALING RESULTS

The results of inference compute scaling on GSM8K are illustrated in Figure 6. Similar to the findings on MATH, Critique-RL is more compute-efficient and significantly increases the performance ceiling, validating the potential of our approach. In addition, we evaluate the refine compute scaling of SFT and Critique-RL across MATH, GSM8K, and AQUA, as illustrated in Figure 7. Critique-RL consistently achieves approximately twice the sampling efficiency of SFT. Notably, with the 7B model on GSM8K, Critique-RL's Pass@1 even surpasses the SFT's Pass@64, demonstrating the effectiveness of our approach.

Table 9: Comparison with other refinement methods with Qwen2.5-3B on GSM8K.

| Method | | GSM8K | |
|---|---|---|---|
| | | **Acc** | **Acc@Dis** |
| Self-Refine | iteration=1 | 71.42 | 75.84 |
| | iteration=2 | 72.71 | 76.52 |
| Critic-CoT | | 70.58 | 74.70 |
| SuperCorrect | | 72.78 | 62.17 |
| Critique-RL (Ours) | | **75.89** | **87.44** |

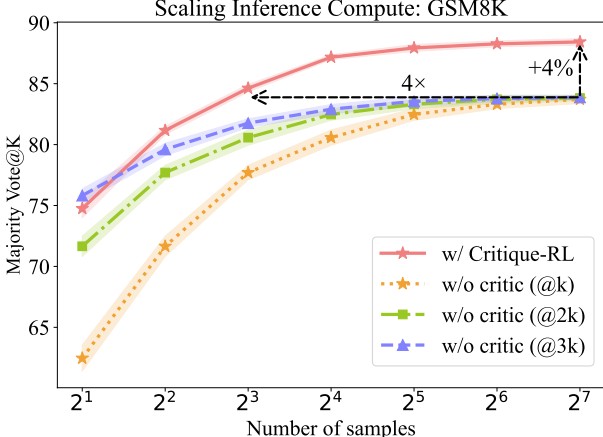

Figure 6: Inference compute scaling for Critique-RL, with @2k and @3k indicating sampling amounts that are 2 times and 3 times the x-axis value, respectively. Critique-RL improves the performance ceiling and is more compute-efficient.

## G PERFORMANCE ON SUMMARIZATION TASK

For open-ended tasks where rule-based verifiers cannot be directly applied, reward signals can be provided through additional reward models or AI feedback (e.g., using GPT-4o (OpenAI, 2023) for judgement).

We conduct experiments of Critique-RL with Qwen2.5-7B-Insturct (Team, 2024) on summarization task using CNN/DailyMail (Hermann et al., 2015) dataset. Specifically, given an article $x$, the actor model generates an original summary $y$. The reward model (Skywork-Reward-V2-Llama-3.1-8B (Liu et al., 2025)) then evaluates the summary, with its output linearly scaled to a 1-10 range, i.e., $r_{\text{oracle}}(x, y)$. Subsequently, the critique model produces critique $c$, which includes comments about the summary across key criteria, a quality score from 1-10, and improvement suggestions. The actor model then generates a revised summary $y'$ accordingly, which is also scored by the reward model to yield a refinement score $r_{\text{refine}} = r_{\text{oracle}}(x, y')$. Based on this, we define the discrimination reward function of the critique model as:

$$r_{\text{dis}}(x, y, c) = \max(0, 1 - \frac{|f(x, y, c) - r_{\text{oracle}}(x, y)|}{\delta})$$

where $f(x, y, c)$ is the quality score of the original summary from critique model. $\delta$ is the permissible maximum error range.

In stage I, we optimize the discriminability of the critique model using $r_{\text{dis}}(x, y, c)$; In stage II, we optimize the helpfulness while maintaining discriminability using the following reward function:

$$r_{\text{stageII}} = r_{\text{refine}} + \beta_1 r_{\text{dis}}(x, y, c)$$

In our experiments, we select 5000 training and 1000 test queries from CNN/DailyMail 3.0.0's official splits. The results are presented in the Table 10.

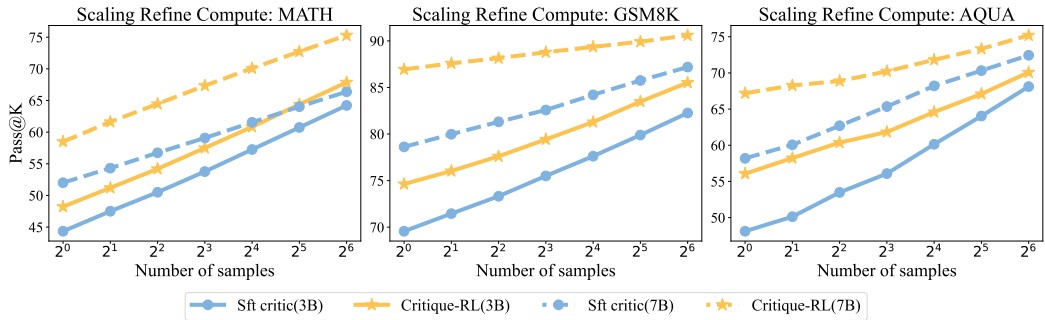

Figure 7: Refine compute scaling for Critique-RL and SFT critic with Qwen2.5-3B and Qwen2.5-7B.

The results reveal that Critique-RL can effectively optimize discriminability, yielding improvement in summary quality. We use MSE and MAE to measure the error between the quality scores produced by the critique model and those from the reward model. Specifically, Critique-RL outperforms baseline by $0.87$ points in Score, $7.87$ points in MSE@Dis and $1.79$ points in MAE@Dis. These improvements

Table 10: Performance on summarization task using Qwen2.5-7B-Instruct. We report the original Score by reward model. The MSE@Dis stands for mean square error, and MAE@Dis stands for mean absolute error, where smaller values indicate stronger discrimination abilities.

| Method | CNN/MD | | | |
| --- | --- | --- | --- | --- |
| | Score↑ | Delta↑ | MSE@Dis↓ | MAE@Dis↓ |
| No Critic | 19.69 | - | - | - |
| 7B-Instruct | 19.94 | 0.25 | 9.46 | 2.77 |
| Critique-RL (Ours) | **20.81** | **1.12** | **1.59** | **0.98** |

demonstrate the strong generalization ability of our approach to open-ended tasks, contributing to scalable oversight.

## H  VALIDATING THE EFFECTIVENESS OF CRITIQUE MODEL

Introducing a separate critique model leads to increased manual effort and additional complexity. To validate the usage of the critique model, we compare Critique-RL with actor-only RL method to show that training a critique model provides significant benefits over directly optimizing the actor. In particular, for actor-only method, we conduct experiments on directly RL the actor and SCoRe (Kumar et al., 2024); for actor-critic paradigm, we use a SFT-based critique model as well as our Crituqe-RL. For a fairer comparison, we train the actor model using the same reasoning traces as Critique-RL in direct RL and using the same reasoning, critique and refinement dataset as Critique-RL in SCoRe. All experiments are conducted with Qwen2.5-7B on the Math dataset.

Table 11: Comparison with actor-only RL method.

| Category | Method | MATH | |
| --- | --- | --- | --- |
| | | **Acc** | **Acc@Dis** |
| Actor-only | Directly RL | 49.78 | - |
| | SCoRe | 56.52 | 72.51 |
| Actor-Critique | SFT | 51.84 | 67.59 |
| | Critique-RL | **58.40** | **85.20** |

The results in Table 11 show that Critique-RL significantly outperforms Directly RL by $8.62$ points in terms of Acc. Also Critique-RL outperforms SCoRe by $12.69$ points in terms of Acc@Dis, and $1.88$ points in terms of Acc. Note that during the training process of Critique-RL, the actor model remained fixed and is thus inherently weaker in reasoning and refinement than the trained SCoRe actor model. Importantly, the trained critique model can be flexibly applied to other stronger actor models (weak-to-strong) and reasoning models to further improve their performance(see Section 6). This modularity and transferability are advantages that SCoRe lacks.

Moreover, we conduct the test-time scaling experiment. The majority vote (MV@K) results are as shown in Table 12. The results show that even the actor model has been well-trained, generating parallel responses still underperforms Critique-RL's response-critique-refinement process. Notably, Critique-RL's MV@1 even surpasses Directly RL's MV@12. This highlights the compute-efficiency of Critique-RL.

Table 12: Performance comparison between Directly RL and Critique-RL under MV@K.

| K | Directly RL | | | Critique-RL |
| --- | --- | --- | --- | --- |
| | MV@K | MV@2K | MV@3K | MV@K |
| 1 | 49.78 | 50.05 | 52.39 | **58.40** |
| 2 | 50.05 | 53.49 | 55.04 | **59.10** |
| 4 | 53.49 | 55.08 | 56.75 | **65.91** |

## I  SENSITIVITY ANALYSIS

For solidness, we provide details about different values for $\beta$, $\beta_1$, $\beta_2$ and training steps per stage.

**Experiments on different values for $\beta$, $\beta_1$, and $\beta_2$.** We exemplify our selection of the parameters $\beta$, $\beta_1$, and $\beta_2$ by presenting the performance of the Qwen2.5-3B model on the GSM8K dataset as an example. The results in Table 13 reveal that these parameters are not sensitive, so we ultimately choose $\beta = 0.01$, $\beta_1 = 0.9$, and $\beta_2 = 0.95$ for our experiments.

**Experiments on different training steps per stage.** We show the performance of the two stages of Critique-RL at different training steps with Qwen2.5-3B on MATH dataset. The results in Table 14 indicate that within 500 steps of Stage I, the model's discriminability was substantially enhanced, with Acc@Dis rising from 66.51 to 78.68. During Stage II, the model maintained this discriminability while further improving helpfulness, with Acc increasing from 45.90 to 48.60.

While further refinement of parameters could potentially yield additional performance gains, the current experimental outcomes are already statistically sound and adequately substantiate our core conclusions.

## J  QUALITATIVE ANALYSIS

We perform a qualitative investigation into how Critique-RL works and provide several examples in Appendix J. In Figure 8, facing the originally incorrect response, the critique model after SFT is unable to detect errors, leading the actor's refinement response to retain the same errors. However, the model trained after Critique-RL identifies the errors in the original response and provides detailed, constructive suggestions for modification, leading to the correct refinement response. In Figure 9, model trained after Critique-RL Stage I is able to detect errors, demonstrating its discriminability. However, the model provides the actor with low-quality suggestion, causing the actor's refinement response to be incorrect. In contrast, for the same erroneous original response, model trained after Critique-RL Stage II not only detects the error but also offers a constructive suggestion, ultimately leading to the correct refinement response, demonstrating the advantage of two-stage RL process.

To directly assess the quality of critiques generated by Critique-RL, we randomly collect 600 critiques that successfully helped refine incorrect answer into correct ones. We leverage GPT-4o with ground-truth answers and solutions as references to evaluate quality more accurately. The results show that 96.2% of these critiques made correct discriminative judgments, and 93.3% were rated as high-quality, demonstrating that Critique-RL produces reliable and helpful critiques.

Table 13: Results of different values for $\beta$, $\beta_1$, and $\beta_2$ with Qwen2.5-3B on GSM8K.

| Parameter | Value | Acc | Delta | Acc@Dis |
|-----------|-------|-------|-------|---------|
| $\beta$ | 0.008 | 74.60 | 8.57 | 86.24 |
| | 0.01 | 75.89 | 9.86 | 87.44 |
| | 0.012 | 74.22 | 8.19 | 87.10 |
| $\beta_1$ | 0.88 | 74.60 | 8.57 | 86.18 |
| | 0.9 | 75.89 | 9.86 | 87.44 |
| | 0.92 | 74.68 | 8.65 | 86.09 |
| $\beta_2$ | 0.93 | 74.68 | 8.65 | 85.99 |
| | 0.95 | 75.89 | 9.86 | 87.44 |
| | 0.97 | 74.37 | 8.34 | 85.74 |

Table 14: Results of different training steps per stage with Qwen2.5-3B on MATH.

| Step | Critique-RL Stage I | | Critique-RL Stage II | |
|------|------|---------|------|---------|
| | Acc | Acc@Dis | Acc | Acc@Dis |
| 0 | 44.24 | 66.51 | 45.90 | 78.68 |
| 100 | 44.22 | 68.26 | 45.88 | 80.56 |
| 200 | 44.60 | 71.53 | 46.82 | 81.77 |
| 300 | 44.89 | 75.72 | 47.02 | 82.47 |
| 400 | 45.18 | 78.20 | 47.90 | **83.06** |
| 500 | **45.90** | **78.68** | **48.60** | 82.80 |

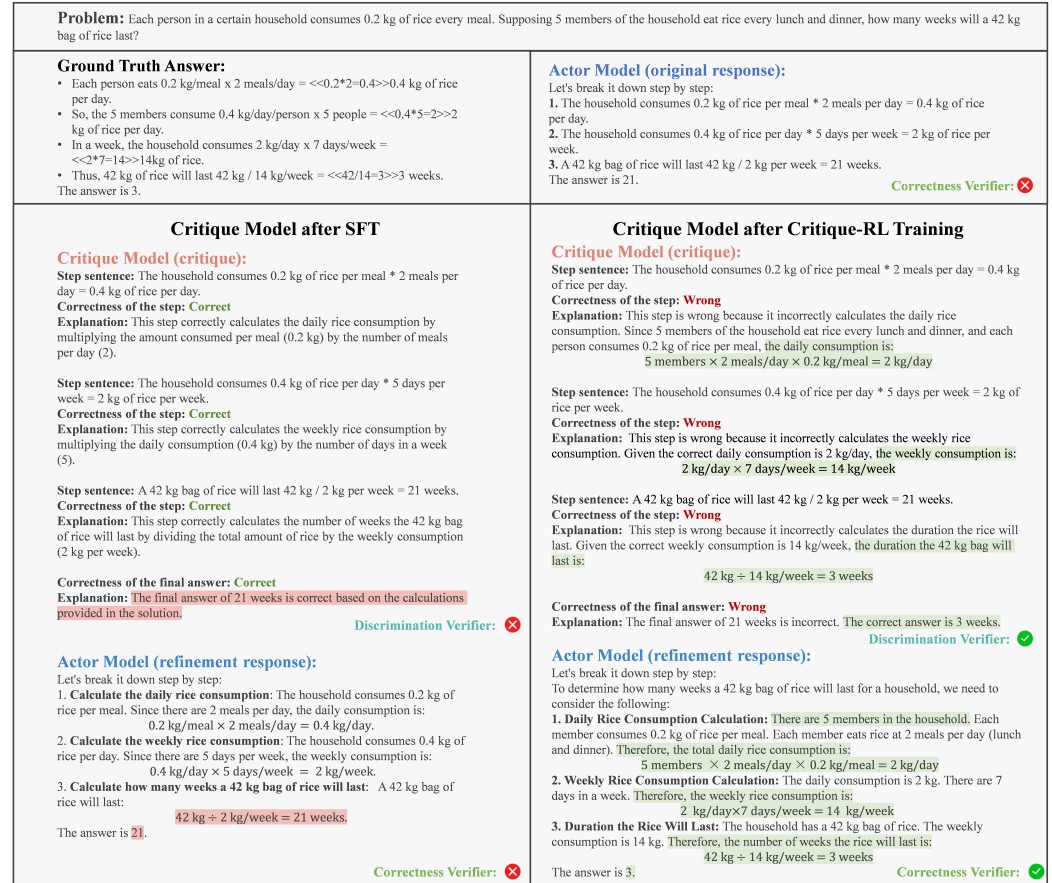

Figure 8: Example 1 of qualitative analysis. The actor's original response is incorrect. The model after SFT is unable to detect errors in the response, leading the actor's refinement response to retain the same errors. However, the model trained after Critique-RL identifies the errors in the original response and provides detailed, constructive suggestions for modification, leading to the correct refinement response.

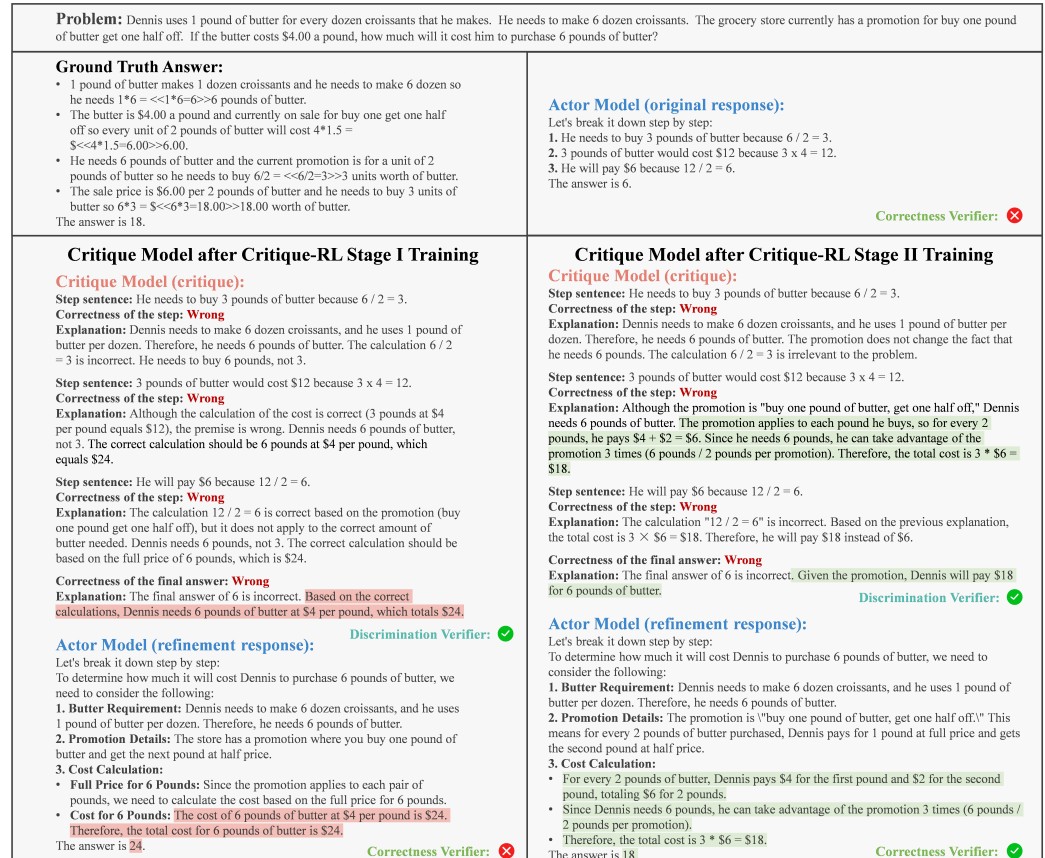

Figure 9: Example 2 of qualitative analysis. The actor's original response is incorrect. The model trained after Critique-RL Stage I is able to detect this error, demonstrating its discriminability. However, the model provides the actor with low-quality suggestion, causing the actor's refinement response to be incorrect. In contrast, for the same erroneous original response, model trained after Critique-RL Stage II not only detects the error but also offers a constructive suggestion, ultimately leading to the correct refinement response, demonstrating the advantage of two-stage RL process.

