# OpenReview forum: "Critique-RL: Training Language Models For Critiquing Through Two-Stage Reinforcement Learning"
_ICLR.cc/2026/Conference — ICLR 2026 Poster_

### Official Review · Reviewer_FEHf · 2025-10-31

**Soundness:** 3
**Presentation:** 3
**Contribution:** 3
**Rating:** 6
**Confidence:** 3

**Summary:**

The paper proposes Critique-RL, a two-stage reinforcement learning (RL) framework for training critiquing language models. The critic  models that assess the correctness of an actor’s response and provide natural language feedback to guide refinement. The core motivation is to avoid reliance on stronger supervisors for critique annotation, which is costly and hard to scale. The key insight is that optimizing solely via indirect rewards, whether the actor’s refined output is correct, leads to poor discriminability, the critic’s ability to accurately judge whether an initial response is correct. To address this, Critique-RL introduces a two-stage training process. Experiments on mathematical reasoning benchmarksshow consistent improvements over baselines.

**Strengths:**

1. The problem setup, motivation, and methodology are clearly articulated with intuitive figures (e.g., Figure 2, 3) and precise definitions of evaluation metrics
2. Scalable oversight remains a critical bottleneck in LLM alignment and self-improvement. By explicitly disentangling and jointly optimizing two key critique capabilities, the work offers a practical and empirically validated pathway toward more reliable critique models.
3. The identification of problem due to conflicting objectives under indirect rewards is compelling.

**Weaknesses:**

1. The paper claims to avoid “stronger supervisors” for critique data, yet relies heavily on ground-truth answers (i.e., golden labels) to construct both direct (Stage I) and indirect (Stage II) reward signals. These labels are typically human-annotated or derived from curated datasets (e.g., MATH, GSM8K). Thus, the method still depends on human-provided supervision, albeit not in the form of natural language critiques. This undermines the central motivation that existing approaches “rely on stronger supervisors for annotating critique data”—since here, human-labeled answers serve as an equally strong (if not stronger) form of supervision.
2. Lack of analysis of some highly related works in this paper [1], which also proposes to use the refinment as the additional feedback.

> [1] Training Language Models to Critique With Multi-agent Feedback

**Questions:**

No question

---

> ### Author Response · Authors · 2025-11-25
> **Response to Reviewer FEHf**
>
> We appreciate your valuable time, insights, and for highlighting our strengths (e.g., **the clear problem setup, motivation, and method; the compelling identification of problem; the effective and practical method**). In the following, we will carefully respond to your questions. We hope we can address your concerns.
>
> ### **1. Question about our motivation.**
>
> Thank you for your question. It made us realize that further clarification is needed.
>
> Some prior works rely on human-annotated critique data to train models via behavior cloning [1, 2]. In contrast, our goal is to eliminate dependence on such annotation pipelines. In our terminology, the stronger human-annotated signals provided by more capable human supervisors correspond to what we referred to as "stronger supervision." This usage follows the notion of "supervision" in supervised learning, where the annotator plays the role of the supervisor.
>
> In our work, the verifier acts as an environment-based reward signal rather than a source of human labels. Our broader vision is to explore whether, in scenarios where humans cannot provide sufficiently strong annotations, a model can nevertheless improve and evolve through environment feedback and self-exploration, ultimately achieving strong generalization on held-out and even out-of-distribution test sets — similar in spirit to the goals of superalignment [3]. **Therefore, while our method still uses verifiers, it fundamentally differs from traditional strong supervised critique data.**
>
> We apologize for the confusion caused by our terminology. We will revise the paper and replace the phrase "stronger supervision" with "stronger labeling" to avoid misunderstanding. Thanks for your question insightful again!
>
> ---
>
> ### **2. Question about comparison with some highly related works.**
>
> Thank you very much for your suggestion. It made us realize that we had not compared our method with this highly relevant prior work. Their approach first utilizes multiple models (GPT-4, Claude, Qwen-72B, etc.) to generate structured, multi-dimensional  Analytical Critique Units (ACUs), which are then aggregated and filtered by GPT-4 to construct a high-quality MultiCritique-SFT dataset. Subsequently, in preference pair construction, samples are selected based on differences in severity scores and further validated for quality through multiple 7B models performing refinements, resulting in reliable preference data used to train a reward model that provides reward signals during RL.
>
> Considering the computational cost and API usage required to fully reproduce this method, we evaluated using their open-sourced MultiCritique-RL-7B as the critique model. The experimental results are as follows.
>
> |       Method        |   MATH    |   GSM8K   |
> | :-----------------: | :-------: | :-------: |
> |                     |    Acc    |    Acc    |
> | MultiCritique-RL-7B |   53.92   |   80.59   |
> |     Critique-RL     | **58.40** | **87.72** |
>
> The experimental results show that Critique-RL outperforms MultiCritique-RL-7B by 4.48 points on MATH and by 7.13 points on GSM8K.
>
> ---
>
> We once again thank you for your insightful comments and suggestions!  If you have any further questions, please feel free to let us know and we will do our utmost to respond. If you find our replies satisfactory, we kindly ask that you consider updating your score and confidence accordingly.
>
> ---
>
> ### **Reference**
>
> [1] Llm critics help catch llm bugs.
>
> [2] Enhancing llm reasoning via critique models with test-time and training-time supervision.
>
> [3] Introducing Superalignment.

---

### Official Review · Reviewer_ao7e · 2025-11-01

**Soundness:** 2
**Presentation:** 3
**Contribution:** 2
**Rating:** 4
**Confidence:** 4

**Summary:**

The paper addresses the challenge of insufficient scalable supervision for LLMs in complex reasoning tasks. It proposes Critique-RL, a two-stage reinforcement learning framework without strong supervision. In the first stage, the critique model’s discriminative ability is optimized using direct rule-based rewards. In the second stage, the framework integrates indirect rewards from agent-corrected response accuracy with regularization to enhance the usefulness of feedback while preserving discriminative capacity. Critique-RL achieves  performance improvements over baselines on several mathematical reasoning and question-answering tasks.

**Strengths:**

- The proposed two-stage RL method is effective to provide constructive critique feedback for better refinement and precise filter for effective time-time scaling.
- The experiments contain several benchmarks across different tasks.
- This paper is well-written and easy to follow.

**Weaknesses:**

My main concern lies in the experimental design, as I am not fully convinced that the current experiments sufficiently demonstrate the proposed method’s advantage on complex reasoning tasks.

- Since the authors explicitly state in the Abstract and Introduction that their method targets complex reasoning tasks, more challenging benchmarks such as AIME and GPQA should have been included in the evaluation.

- Although the main text reports significant improvements on Qwen-3B and Qwen-7B, the appendix reveals that the performance gains on stronger models, such as DeepSeek-R1-Distill-Qwen-7B and Qwen2.5-72B-Instruct, are quite limited. These results should be reported and discussed in the main paper.

- The proposed method requires access to ground-truth answers to compute rewards. Under this setting, it remains unclear what advantages it offers over directly training the model’s reasoning ability using RLVR methods (e.g., GRPO, DAPO). Additional experiments are needed to clarify this distinction.

**Questions:**

See weakness

---

> ### Author Response · Authors · 2025-11-25
> **Response to Reviewer ao7e. Part [1/2]**
>
> We appreciate your valuable time, insights, and for highlighting our strengths (e.g., **the method and the diverse evaluation tasks**). In the following, we will carefully respond to your questions. We hope we can address your concerns.
>
> ### **1. Suggestion about including more challenging benchmarks like AIME and GPQA.**
>
> Thank you for this suggestion, which made us realize that including more challenging benchmarks would further strengthen the reliability of our paper's conclusions. Following your suggestion, we trained using Qwen2.5-7B-Instruct and evaluated on the AIME2024, AIME2025 [1], and GPQA-Diamond [2] benchmarks. We used General-Reasoner-7B [3] as the actor and constructed a training set of 30k examples based on the WebInstruct-Verified[3] dataset for RL training. The experimental results are as follows:
>
> | **Methods** | **GPQA-Diamond** |          |           | **AIME2024** |          |           | **AIME2025** |          |           |
> | :---------: | :--------------: | :------: | :-------: | :----------: | :------: | :-------: | :----------: | :------: | :-------: |
> |             |       ACC        |  Delta   |  ACC@Dis  |     ACC      |  Delta   |  ACC@Dis  |     ACC      |  Delta   |  ACC@Dis  |
> |  No Critic  |      34.30       |    -     |     -     |    11.98     |    -     |     -     |     6.67     |    -     |     -     |
> |     SFT     |      35.86       |   1.56   |   41.20   |    12.30     |   0.32   |   22.71   |     7.50     |   0.83   |   6.67    |
> | Critique-RL |    **37.37**     | **3.07** | **51.52** |  **13.75**   | **1.77** | **53.44** |   **8.50**   | **1.83** | **30.10** |
>
> The experimental results demonstrate that our method significantly improves the critique model's discriminability, with particularly notable improvements on the challenging reasoning datasets AIME2024 and AIME2025. Without fine-grained hyperparameter adjustments, our method outperforms the baseline across all three benchmarks, proving the effectiveness of Critique-RL in complex reasoning scenarios. We have now included the experimental results and corresponding discussion in Appendix C of the revised manuscript.
>
> ---
>
> ### **2. Suggestion about discussing results on stronger models.**
>
> Thank you for this comment. We have supplemented the results for Qwen2.5-72B-Instruct and DeepSeek-R1-Distill-Qwen-7B in the main text (Section 6) with discussion. **It is important to emphasize that the experiments in the paper use a 7B-scale critique model.** Therefore, when critiquing more powerful actors (such as 72B or R1-Distill-7B), the overall improvement potential is limited by the capability gap between models, resulting in relatively modest performance gains.
>
> To better illustrate this point, we trained Qwen2.5-72B-Instruct using Critique-RL and used it as the critique model for re-evaluation. The results are shown in the following table:
>
> |               Methods               | In-Domain: MATH-500 |       | OOD: Theoremqa |       |
> | :---------------------------------: | :-----------------: | :---: | :------------: | :---: |
> |                                     |         Acc         | Delta |      Acc       | Delta |
> |             Base Model              |        79.10        |   -   |     21.38      |   -   |
> | Critique-RL with 7B Critique Model  |        80.30        | 1.20  |     23.50      | 2.10  |
> | Critique-RL with 72B Critique Model |        82.00        | 2.90  |     25.25      | 3.87  |
>
> The experimental results show that the trained 72B-Instruct brings a performance improvement of 2.9 points within domain, 3.87 points to the actor in out-of-domain tasks, respectively. When the critic and actor are at the same capability level, Critique-RL can provide considerable improvements. Therefore, the limited gains when using a 7B critic primarily stem from the model scale mismatch rather than limitations of the Critique-RL method itself.

---

> ### Author Response · Authors · 2025-11-25
> **Response to Reviewer ao7e. Part [2/2]**
>
> ### **3. Question about comparison with RLVR methods.**
>
> Thank you for your valuable feedback! We would kindly like to point out that we have already compared Critique-RL with directly training the actor using GRPO in Appendix H, Tables 10 and 11.
>
> The results in Table 10 show that Critique-RL significantly outperforms Directly RL the actor by 8.62 points in terms of Acc. Also Critique-RL outperforms SCoRe [4] by 12.69 points in terms of Acc@Dis, and 1.88 points in terms of Acc. Note that during the training process of Critique-RL, the actor model remained fixed and is thus inherently weaker in reasoning and refinement than the trained SCoRe actor model. Importantly, the trained critique model can be flexibly applied to other stronger actor models (weak-to-strong) and reasoning models to further improve their performance(see Appendix C). This modularity and transferability are advantages that SCoRe lacks. Moreover, we conduct the test-time scaling experiment.
>
> |    Category    |   Method    |   MATH    |           |
> | :------------: | :---------: | :-------: | :-------: |
> |                |             |    Acc    |  Acc@Dis  |
> |   Actor-only   | Directly RL |   49.78   |     -     |
> |                |    SCoRe    |   56.52   |   72.51   |
> | Actor-Critique |     SFT     |   51.84   |   67.59   |
> |                | Critique-RL | **58.40** | **85.20** |
>
>
> The majority vote (MV@K) results are as shown in Table 11. The results show that even the actor model has been well-trained, generating parallel responses still underperforms Critique-RL's response-critique-refinement process. Notably, Critique-RL's MV@1 even surpasses Directly RL's MV@12. This highlights the effectiveness of Critique-RL in the test-time scaling setting.
>
> |      | Directly RL MV@K | Directly RL MV@2K | Directly RL MV@3K | Critique-RL MV@K |
> | ---- | :--------------: | :---------------: | :---------------: | :--------------: |
> | K=1  |      49.78       |       50.05       |       52.39       |    **58.40**     |
> | K=2  |      50.05       |       53.49       |       55.04       |    **59.10**     |
> | K=4  |      53.49       |       55.08       |       56.75       |    **65.91**     |
>
> ---
>
> We once again thank you for your insightful comments and suggestions!  If you have any further questions, please feel free to let us know and we will do our utmost to respond. If you find our replies satisfactory, we kindly ask that you consider updating your score accordingly.
>
> ---
>
> ### **References**
>
> [1] AIME. Aime problems and solution, 2025.
>
> [2] GPQA: A graduate-level google-proof q&a benchmark.
>
> [3] General-Reasoner: Advancing LLM Reasoning Across All Domains.
>
> [4] Training Language Models to Self-Correct via Reinforcement Learning.

---

> > ### Comment · Reviewer_ao7e · 2025-11-27
> > **Response to author**
> >
> > Thank you for the response. However, I am quite confused about the experiments involving RLVR.
> >
> > 1. Questions regarding the setup of Table 10: What is the base model, evaluate settings, training data and exact training procedure used in these experiments?
> > And why does Table 10 not include benchmarks on  GSM-8K, GPQA, AIME, etc.?
> >
> > 2. Concerns about the reported results: I am particularly surprised by the results in Table 10, where the critique-RL variant appears to significantly outperform RLVR. Based on both public reports and my own experimental experience, the performance of critique-RL (base QWEN-2.5-7B) in this paper **does not** outperform RLVR approaches such as GRPO or DAPO on GSM-8K, AIME, GPQA, etc. (especially AIME).
> > Some of these results are summarized in the official VERL report (only baseline):
> > https://verl.readthedocs.io/en/latest/algo/baseline.html

---

> > > ### Author Response · Authors · 2025-11-28
> > > **New response to Reviewer ao7e**
> > >
> > > Thank you for your valuable and constructive feedback, as well as your timely engagement during the rebuttal phase. Below, we carefully address your questions, and we hope our clarifications can resolve your concerns.
> > >
> > > ### **1.Question regarding the setup in RLVR methods**
> > >
> > > In Table 10, we perform SFT on **Qwen2.5-7B-Base** using the **train split of the MATH dataset** (7,500 samples; link: https://huggingface.co/datasets/DigitalLearningGmbH/MATH-lighteval), which contains both queries and trajectories. We train for 3 epochs with a learning rate of 5e-6. After SFT, we apply GRPO to this SFT-initialized **Qwen2.5-7B-Base**. For evaluation, we use greedy decoding with a maximum generation length of 1024 tokens.
> > >
> > > We did not test on other datasets because we only trained on the MATH training split, which **does not contain modern long-chain-of-thought annotations** and is not a multi-domain SFT dataset. Therefore, we did not include AIME and GPQA in the evaluation. Later, following your suggestion, to ensure more comprehensive evaluation, we instead used a **stronger and multi-domain actor model, General-Reasoner-7B**, which is trained on multi-domain RL  datasets via Zero-RL [1].
> > >
> > > ### **2. Concerns about the results in RLVR methods**
> > >
> > > As explained above, Table 10 is based on the **original MATH** **training set**, with generation  length limited to 1024. This dataset is **not** a modern long-CoT dataset (e.g., OpenMathReasoning). Additionally, our RL is performed on **Qwen2.5-7B-Base**, whereas the RLVR baselines you referenced (https://verl.readthedocs.io/en/latest/algo/baseline.html) are trained using **Qwen2.5-7B-Instruct**, which already possesses substantially stronger reasoning and instruction-following capability.
> > >
> > > For this reason, i**n the rebuttal** we further experiment with stronger models and broader datasets to evaluate performance on AIME and GPQA. Specifically, we use the **General-Reasoner-7B**[1]—a model obtained by applying Zero-RL to Qwen2.5-7B-Base, and reported to achieve strong multi-domain reasoning performance (not limited to mathematics). According to its paper, its training context length is 4096. Using General-Reasoner-7B as the actor, we retrained our critique models and evaluated them on AIME 2024, AIME 2025, and GPQA. The results verify that our method continues to provide measurable improvements even when the actor has already been strengthened through RLVR training. This supports our goal of **scalable oversight**: the trained critique model can provide useful supervision across domains (both in-domain and out-of-domain) and for actors of varying capability, whether or not they have been trained with RLVR methods.
> > >
> > > [1] General-Reasoner: Advancing LLM Reasoning Across All Domains.

---

> > > > ### Comment · Reviewer_ao7e · 2025-11-28
> > > >
> > > > Thank you for your reply, I will raise my score. At the same time, I suggest adding common RLVR algorithms as baselines in the main experiments of the paper.

---

> > > > > ### Author Response · Authors · 2025-12-02
> > > > > **Thank you!**
> > > > >
> > > > > Dear Reviewer ao7e,
> > > > >
> > > > > We sincerely appreciate your valuable feedback, thoughtful engagement throughout the rebuttal phase, and your positive reconsideration of our work. We will continue to further improve our work!
> > > > >
> > > > > Best regards,
> > > > >
> > > > > Authors of ICLR 2026 Submission 20154

---

### Official Review · Reviewer_WXJP · 2025-11-08

**Soundness:** 3
**Presentation:** 3
**Contribution:** 2
**Rating:** 4
**Confidence:** 4

**Summary:**

This paper proposes Critique-RL, a two-stage reinforcement learning (RL) approach to train critiquing language models without requiring stronger supervision. The authors first show that baseline RL methods, which use only indirect rewards from an actor's refinement, fail. This is because they improve the critic's helpfulness (constructive feedback) but not its discriminability (judging correctness), leading to poor performance. Critique-RL solves this by:
- Stage I: Explicitly optimizing discriminability using direct, rule-based reward signals.
- Stage II: Optimizing helpfulness using indirect rewards (actor refinement) while using regularization to maintain the discriminability from Stage I.

This two-stage strategy delivers performance improvements on both in-domain and out-of-domain tasks, e.g., +9.02% in-domain and +5.70% OOD for Qwen2.5-7B.

**Strengths:**

- The paper's core originality is its clear diagnosis of a key failure mode in training critics: baseline RL methods create a conflict between "discriminability" (judging correctness) and "helpfulness" (providing feedback), optimizing the latter at the expense of the former.
- The paper's quality is high, with a rigorous methodology. The training dynamics in Figure 3 clearly show the baseline's failure , while decisive ablation studies in Table 3 prove that both stages of Critique-RL and its specific regularization are essential for success.
- The work significantly contributes to scalable oversight by providing an effective method to train critics without stronger supervisors. Its value is shown through broad applicability, including "weak-to-strong" generalization (a 7B critic improving a 72B actor) and effectiveness on OOD and open-ended tasks .

**Weaknesses:**

- The paper's primary motivation is to train critics "without stronger supervision"1. However, the entire method, especially the critical Stage I, is heavily reliant on an "oracle reward function" $r_{oracle}(x,y)$ to compute the direct discrimination reward $r_{dis}$. For the main experiments on math tasks, this oracle is a rule-based verifier that knows the correct answer. This oracle is a form of strong, external supervision.
- The framework's success, particularly in Stage II, hinges on a critical assumption: the fixed actor model $\pi_{\theta}$ is already a good "refiner". The authors state the actor is pre-trained to be "capable of... faithfully refining them according to critiques". This assumes away a large part of the problem. The helpfulness reward $r_{refine}$ is a convolved signal of both the critique's quality and the actor's ability to understand it. If the actor is a poor refiner, $r_{refine}$ becomes a noisy or meaningless signal, and Stage II would fail to optimize for helpfulness.
- The paper correctly highlights its inference-time compute-efficiency benefits (e.g., in Figure 1 and Figure 6). However, it completely omits the training-time cost of this complex, two-stage RL pipeline. Stage II, for example, requires at least three model forward passes per training sample (one for the critic $c=\pi_{\phi}(x,y)$, one for the actor's refinement $y^{\prime}=\pi_{\theta}(x,y,c)$, and one for the oracle/RM $r_{refine}=r_{oracle}(x,y^{\prime})$). This is significantly more expensive than the SFT or baseline RL methods it's compared against.
- The paper's core insight is that final-outcome rewards are insufficient, and a more direct signal is needed. The proposed solution, $r_{dis}$, is a direct reward for judging the outcome of the original response. This overlooks a more direct comparison to Process-based Reward Models (PRMs), which provide supervision at each step of the reasoning. The qualitative examples (Figs. 8, 9) show the critic is evaluating step-by-step, but it is only trained on the final answer's correctness.

**Questions:**

The paper is motivated as an approach for training critique models "without stronger supervision". However, the method's crucial first stage relies entirely on a direct reward $r_{dis}$ from an "oracle reward function" $r_{oracle}(x,y)$. This oracle, which knows the ground-truth correctness, seems to be a form of strong supervision.

Could you please clarify this apparent contradiction?

1. How do you formally define the "stronger supervision" (which you avoid) versus the "oracle verifier" (which you use)?
2. The paper's contribution seems to be a novel way to distill the knowledge of a "weak" binary oracle (answer-checker) into a "strong" generative critic (feedback-generator). Would you agree with this framing?
3. How does this framework scale to complex domains (e.g., creative writing, complex coding) where no such simple oracle or high-quality reward model exists?

---

> ### Author Response · Authors · 2025-11-25
> **Response to Reviewer WXJP. Part [1/4]**
>
> We appreciate your valuable time, insights, and for highlighting our strengths (e.g., **the motivation, and the importance of the problem, and effectiveness of our method**). In the following, we will carefully respond to your questions. We hope we can address your concerns.
>
> ### **1. Question about the motivation and the rule-based verifier.**
>
> Thank you for your question. It made us realize that further clarification is needed. The setting we evaluate focuses on training critique models for scalable oversight and validating the feasibility of this direction.
>
> Some prior works rely on human-annotated critique data to train models via behavior cloning [1][2]. In contrast, our goal is to eliminate dependence on such annotation pipelines. In our terminology, the stronger human-annotated signals provided by more capable human supervisors correspond to what we referred to as "stronger supervision". This usage follows the notion of "supervision" in supervised learning, where the annotator plays the role of the supervisor.
>
> **In our work, the verifier acts as an environment-feedback-based reward signal rather than a source of human annotations.** Our broader vision is to explore whether, in scenarios where humans cannot provide sufficiently strong annotations, a model can nevertheless improve and evolve through environment feedback and self-exploration, ultimately achieving strong generalization on held-out and even out-of-distribution test sets — similar in spirit to the goals of superalignment [3]. Notably, Critique-RL demonstrates improved performance in both in-domain and out-of-domainm, as shown in Table 1 and Table 4 in the paper.
>
> We apologize for the confusion caused by our terminology. We will revise the paper and replace the phrase "stronger supervision" with "stronger labeling" to avoid misunderstanding.

---

> ### Author Response · Authors · 2025-11-25
> **Response to Reviewer WXJP. Part [2/4]**
>
> ### **2. Suggestion about how the framework scale to complex domains.**
>
> Thank you for this suggestion. In our paper, we validate the feasibility of our method on mathematical tasks in the main text and further demonstrate its effectiveness on summarization tasks in the Appendix G. We respond to your question from two perspectives.
>
> **(1) Clarification of the Discrimination Concept and Its Applicability to Complex Tasks**
>
> In the mathematical task experiments in the main text, we define discrimination as "the critique model's ability to judge whether the initial response is correct," and we use an oracle verifier to verify the correctness of initial responses to construct rewards. However, we want to clarify that this does not mean the concept can only be applied to the narrow scenario of "judging correctness."
>
> In more general tasks, we broadly define discrimination as: **the critique model's ability to identify the quality level of initial responses**. As long as a task has a reasonable evaluation benchmark and an operational assessment method, we can define discrimination accordingly and further use it for Critique-RL optimization.
>
> To strengthen this point, we conducted additional experiments using a 7B-Instruct model on the Creative Writing V3 Benchmark [4]. Most evaluation benchmarks for creative writing tasks are based on rubric-based judgment [4][5][6]. Therefore, in our setup, we require the critique model to output a score, scoring reason, and specific improvement suggestions for the given criteria after receiving the query and initial response.
>
> We collected 286 creative writing examples from WritingBench [6] and HelloBench [5] as the training set, and used Writing-Model-Qwen-7B [6] as the actor. The scores provided by WritingBench-Critic-Model-Qwen-7B [6] according to the rubric serve as the source of $r_\text{oracle}$ and $r_\text{refine}$, while we still use $r_\text{dis}$ as defined in Appendix G to provide rewards for training discriminability.
>
> The experimental results are shown in the following table:
>
> | Methods     | Creative Writing V3 |          |           |           |
> | ----------- | ------------------- | -------- | --------- | --------- |
> |             | Score↑              | Delta↑   | MSE@Dis ↓ | MAE@Dis ↓ |
> | No Critic   | 54.72               | -        | -         | -         |
> | SFT         | 55.87               | 1.15     | 1.499     | 3.501     |
> | Critique-RL | **58.54**           | **3.82** | **1.053** | **1.862** |
>
> The experimental results demonstrate that Critique-RL effectively improves the critique model's discrimination ability while bringing significant improvements in writing quality. **This proves the generalizability of our method to complex tasks.**
>
> **(2) Applicability of Our Method When High-Quality Reward Models Are Unavailable**
>
> We agree that in some complex domains, it is difficult to obtain reliable oracles or high-quality reward models, which may reduce the reward quality of $r_\text{oracle}$ and $r_\text{refine}$. To address your concern, we conducted robustness experiments on the MATH task, where the rewards from $r_\text{oracle}$ and  $r_\text{refine}$ have a 10\% probability of being randomly perturbed to simulate the realistic situation of "unreliable supervision".
>
> | Methods               |   MATH    |           |           |
> | --------------------- | :-------: | :-------: | :-------: |
> |                       |    ACC    |   Delta   |  ACC@Dis  |
> | No Critic             |   45.74   |     -     |     -     |
> | SFT                   |   51.84   |   6.10    |     -     |
> | Critique-RL w/ noise  |   56.10   |   10.36   |   78.50   |
> | Critique-RL wo/ noise | **58.40** | **12.66** | **85.20** |
>
> The experimental results show that even under this setting, our method still brings significant performance improvements over SFT. This demonstrates that **our method has a certain degree of robustness** and can still achieve good results when oracle verifiers and high-quality reward models are unavailable.

---

> ### Author Response · Authors · 2025-11-25
> **Response to Reviewer WXJP. Part [3/4]**
>
> ### **3. Question about the refinement model.**
>
> First, we would like to clarify that the actor/refinement model used in Stage I and Stage II is the same model; there is no architectural or parameter difference between them.
>
> Second, while our formulation assumes that the model is faithful, we agree that this does not always hold in practice, and we have observed this issue as well. This is why we conducted the two-stage training: the indirect reward signal can be noisy, and we aim to mitigate this effect. Accordingly, the phrasing in our paper — "capable of … faithfully refining them according to critiques" — may be misleading. We will revise it to "capable of reasonably following critiques." In addition, we performed a manual analysis by sampling 200 examples, and found that the model’s faithfulness rate is approximately 85.5\%.
>
> Furthermore, to better address this concern, we also conducted experiments with injected noise, where the reward signal for the critique model is perturbed with a controlled amount of random noise. **As shown in our second response to you, even under noisy feedback, the critique model’s generalization ability combined with our training strategy still yields strong performance.**
>
> ---
>
> ### **4. Question about the computing efficiency of our method.**
>
> Thank you for this important question! Our goal is to train a critique model with scalable oversight capability from scratch, without relying on heavily supervised annotation data. To achieve strong performance under this constraint, additional computational cost during training is required.
>
> During training, the helpfulness of critiques cannot be directly evaluated using rule-based verifiers. Instead, many previous works such as CTRL [7] and Retroformer [8] assess it through the actor's refinement results: the critic generates critiques, the actor refines based on these critiques, and an oracle verifier or reward model evaluates the refined output. This naturally introduces three forward passes during training. Even if we replace the oracle verifier with a reward model [9], the reward model itself still requires training on critique preference pairs constructed through actor refinement, which does not reduce the overall training cost.
>
> More importantly, introducing additional training compute is a common and necessary practice in LLM reasoning and scalable oversight research. For example:
>
> - **ScoRE [10]** adopts a two-stage training approach to obtain models with self-correction capability. The second stage requires the model to perform two complete inference passes for reasoning and self-correction.
>
> - **CTRL [7] and Retroformer [8]** train critique models capable of providing high-quality critiques through RL, where the actor refines outputs based on critic feedback and an oracle verifier or reward model evaluates the results. This introduces three forward passes during training.
>
> - **Debate[11]** requires models to play "pro" and "con" roles across multiple rounds, continuously proposing, questioning, and refuting reasoning chains, with human judges making final decisions based on the debate. This protocol requires adversarial incentives and multi-round, multi-role reasoning, resulting in significantly higher compute than standard RL.
>
> - **Iterated Amplification[12]** uses multiple weak models to collaboratively perform task decomposition, information completion, and judgment aggregation (Amplify), then distills these aggregated decisions into the main model (Distill). Each training iteration requires executing the full pipeline of "task decomposition → multi-model collaborative reasoning → result aggregation → distillation," introducing substantial computational overhead through multiple forward passes and cross-model interactions.
>
> In conclusion, our two-stage Critique-RL does introduce additional compute during training, but its cost remains on the same order of magnitude as commonly used scalable oversight frameworks in this research area—it is not abnormally inflated. Crucially, once training is complete, our method maintains **high computational efficiency during inference** (as shown in Figures 1 and 6), which is a key advantage of our approach.

---

> ### Author Response · Authors · 2025-11-25
> **Response to Reviewer WXJP. Part [4/4]**
>
> ### **5. Suggestion about comparison to process-based reward models.**
>
> Thank you for this suggestion. Following your recommendation, we have integrated PRMs into the Critique-RL training process. Specifically, we used the original RL dataset and employed DeepSeek-R1 to segment the actor's CoT into steps, inserting step identifiers and verifying the correctness of each step. During RL training, the critique model makes correctness judgments for each step as demarcated by the step identifiers, and we verify these step-level judgments. We then calculate the proportion of steps judged correctly as the process-supervised reward $r_\text{dis}$. The experimental results are as follows:
>
> | Methods           | MATH  |       |         |
> | ----------------- | :---: | :---: | :-----: |
> |                   |  ACC  | Delta | ACC@Dis |
> | No Critic         | 45.74 |   -   |    -    |
> | SFT               | 51.84 | 6.10  |  67.59  |
> | Critique-RL-PRM   | 58.80 | 13.06 |  82.68  |
> | Critique-RL(Ours) | 58.40 | 12.66 |  85.20  |
>
> The results show that Critique-RL-PRM also significantly outperforms SFT, achieving a performance improvement of 6.96 points, slightly surpassing our original Critique-RL method. We observe that the PRM-based approach leads to some degradation in discrimination capability. This may be because the model needs to judge both intermediate steps and final outcomes, making it difficult to achieve an optimal balance between the two. But the performance improvement is still achieved because it more precisely identifies the judgement issue. **However, this approach requires pre-provided step-level verification, which introduces additional cost.** In contrast, our method does not rely on step-level supervision signals and can construct $r_\text{dis}$ using only the final outcome.
>
> ---
>
> ### **6. Question about definition of stronger supervision and the oracle verifier.**
>
> Thanks for the important question! Please refer to our first response to you.
>
> ---
>
> ### **7. Question about whether our method is a novel way to distill the knowledge of a “weak” binary oracle (answer-checker) into a “strong” generative critic (feedback-generator).**
>
> We **do not fully** agree with this interpretation. Our goal is to study a scalable oversight setting: we aim to train a model on the training set where an answer-checker is available, such that at deployment time—where no answer-checker exists—and even on OOD test sets, the model can still provide effective supervision and useful feedback to improve the actor’s performance.
>
> First, we acknowledge that our method does distill part of the binary oracle verifier's ability on the training set, primarily regarding answer correctness. **However, this verifier (i) does not know where the error occurs,** at which step it happens, or what the underlying cause is (as shown in Figure 9 of the paper), and **(ii) provides no natural-language critique or step-level information** that can be "distilled" into actionable feedback. It can only instruct the actor to re-rollout in a large sampling space, offering no guidance on how to improve.
>
> Furthermore, on the test set and especially OOD data where no oracle verifier is available at all, the critique model becomes essential.
>
> ---
>
> We once again thank you for your valuable comments!  If you have any further questions, please feel free to let us know and we will do our utmost to respond. If you find our replies satisfactory, we kindly ask that you consider updating your score accordingly.
>
> ---
>
> ### **References**
>
> [1] Llm critics help catch llm bugs.
>
> [2] Enhancing llm reasoning via critique models with test-time and training-time supervision.
>
> [3] Introducing Superalignment.
>
> [4] Eq-bench: An emotional intelligence benchmark for large language models.
>
> [5] Hellobench: Evaluating long text generation capabilities of large language models.
>
> [6] Writingbench: A comprehensive benchmark for generative writing.
>
> [7] Teaching Language Models to Critique via Reinforcement Learning.
>
> [8] Retroformer: Retrospective Large Language Agents with Policy Gradient Optimization.
>
> [9] Training Language Models to Critique With Multi-agent Feedback.
>
> [10] Training Language Models to Self-Correct via Reinforcement Learning.
>
> [11] AI safety via debate.
>
> [12] Supervising strong learners by amplifying weak experts.

---

### Official Review · Reviewer_4UWL · 2025-11-08

**Soundness:** 3
**Presentation:** 3
**Contribution:** 3
**Rating:** 6
**Confidence:** 4

**Summary:**

This paper proposes an online RL approach called Critique-RL for developing critiquing language models without stronger supervision. This approach contains a two-player paradigm, where the actor generates a response, the critic provides feedback, and the actor refines the response accordingly. The authors devise a two-stage optimization strategy, where stage I reinforces the discriminability of the critic with direct rule-based reward signals and stage II introduces indirect rewards based on actor refinement to improve the critic’s helpfulness. Experimental results show the effectiveness of Critique-RL.

**Strengths:**

1. The proposed two-stage RL method is sound and well-motivated, which deals with the core problem of critique generation.
2. Extensive experiments show the effectiveness of the proposed method.
3. This paper is overall well-written and easy to follow.

**Weaknesses:**

1. The design of indirect rewards based on actor refinement is similar to [1], which is not discussed in the current paper. The authors should further clarify the difference between this work and [1] to highlight their novelty.

2. The quality of generated critiques should be individually measured via automatic metrics or human evaluation.


[1] Training Language Model to Critique for Better Refinement. ACL 2025 Findings.

**Questions:**

I have included my questions in the weaknesses part.

---

> ### Author Response · Authors · 2025-11-25
> **Response to Reviewer 4UWL**
>
> We appreciate your valuable time, insights, and for highlighting our strengths (e.g., **the motivation, and the importance of the problem, and effectiveness of our method**). In the following, we will carefully respond to your questions. We hope we can address your concerns.
>
> ### **1. Suggestion about comparison with "Training Language Model to Critique for Better Refinement."**
>
> Thank you very much for the suggestion. This pointed out that we had not compared against a highly relevant prior work. We have now conducted experiments using Qwen-2.5-7B to reproduce this baseline, RCO (Refinement-oriented Critique Optimization). The results show that RCO performs quite well, but it still does not surpass our method. We believe this may be because RCO does not involve online RL training and does not explicitly optimize the model’s judgment capabilities. In future work, we plan to incorporate RCO into an online RL training framework and examine its effects. We expect this to provide further insights.
>
> |     Method      | MATH  | GSM8K |
> | :-------------: | :---: | :---: |
> |     **RCO**     | 55.16 | 81.43 |
> | **Critique-RL** | 58.40 | 87.72 |
>
> ---
>
> ### **2. Suggestion about evaluating critiques individually.**
>
> Thank you very much for your helpful suggestion and kind feedback. We would like to respectfully clarify that we have already conducted the relevant analysis in Appendix J. Specifically, We randomly collect 600 critiques that successfully helped refine incorrect answer into correct ones. We leverage GPT-4o with ground-truth answers and solutions as references to evaluate quality more accurately. The results show that 96.2% of these critiques made correct discriminative judgments, and 93.3% were rated as high-quality, demonstrating that Critique-RL produces reliable and helpful critiques.
>
> To further validate your concern, we additionally recruited three volunteers to evaluate 300 samples using the same protocol. We found that more than 95.0% of the critiques made correct judgments, and 92.3% of the samples were assessed as providing high-quality feedback. These results further support the reliability and effectiveness of our method.
>
> ---
>
> We once again thank you for your valuable comments!  If you have any further questions, please feel free to let us know and we will do our utmost to respond. If you find our replies satisfactory, we would greatly appreciate it if you could update your score accordingly.

---

### Author Response · Authors · 2025-12-02
**Summary**

We sincerely thank the reviewers, the Area Chair, the Senior Area Chairs and Program Chairs for the valuable feedback. We also deeply value thoughtful evaluations, which highlight several strengths of our works:

1. Well-motivated and sound two-stage RL framework that targets the core failure mode of critique training by disentangling discriminability and helpfulness. (Reviewer 4UWL, WXJP, ao7e, FEHf)
2. Clear identification and thorough analysis of unsatisfactory optimization under indirect rewards, significantly contributes to scalable oversight. (Reviewer WXJP, FEHf)
3. Strong empirical validation through extensive experiments and decisive ablations across multiple benchmarks and task types. (Reviewer 4UWL, WXJP, ao7e)
4. Demonstrated scalability and broad applicability, including weak-to-strong generalization, OOD generalization, and effectiveness on open-ended tasks. (Reviewer WXJP)
5. High-quality presentation with rigorous methodology, intuitive figures, and clear, well-organized writing that is easy to follow. (Reviewer 4UWL, WXJP, ao7e, FEHf)

A major concern shared by reviewers is the role of stronger supervision and the oracle verifier. Another concern is insufficient refinement capability of the actor may prevent effective optimization of helpfulness. A common suggestion across reviewers is to compare Critique-RL with similar methods from recent papers. Reviewers also highlighted the need for Critique-RL’s performance on challenging benchmarks. Additionally, integrating PRMs that supervise judgments at each step into Critique-RL and scaling to more complex domains where high-quality reward model are unavailable were suggested.

In our rebuttal, we conduct additional experiments and offer clear analysis to address these questions, including:

1. Comparison with RCO and MultiCritique methods mentioned by reviewer.
2. Generalization of Critique-RL to creative writing tasks and challenging benchmarks such as AIME and GPQA.
3. Robustness analysis under noisy reward settings where rewards have a probability of being randomly perturbed.
4. Comparison with methods integrating PRMs into the Critique-RL training process, showing that Critique-RL achieves comparable performance with lower supervision requirements.
5. Evaluation of Critique-RL trained with a 72B critic to demonstrate the limited improvement in weak-to-strong settings (7B critic to 72B actor) is due to model capacity mismatch rather than method limitations.
6. Clarification of the role of stronger supervision and oracle verifiers, as well as the computational cost involved in Critique-RL training.

All the above results are consistently promising, offering additional validation of the Critique-RL method's effectiveness and generalizability.

**Our rebuttal received positive recognition from Reviewer ao7e, who raised the overall score.** Although other reviewers did not participate in the discussion, we made every effort to address their concerns through detailed responses and additional experiments.

We have revised our manuscript based on the rebuttal content to further strengthen our work. Finally, we greatly appreciate the reviewers' recognition of Critique-RL and their constructive suggestions. We believe that our work can provide insights for the scalable oversight community of large language models.

---

### Meta-Review · Area_Chair_NAHb · 2026-01-07

**Summary:**

The reviewers raised concerns primarily regarding the paper’s novelty and positioning relative to closely related refinement-based works (e.g., RCO, MultiCritique-RL), the reliance on oracle/verifier signals that could be interpreted as strong supervision, the generalizability of the method beyond mathematical reasoning tasks to complex domains (e.g., creative writing, AIME, GPQA), the computational efficiency and practical cost of the two-stage RL training framework, and the completeness and clarity of the experimental evaluation across stronger actor models. While several reviewers acknowledged the effectiveness of Critique-RL and its practical significance for scalable oversight, there was concern that the original submission did not fully establish a sufficiently distinct or rigorously supported contribution.

**Reviewer Concerns:**

Many of the reviewers’ core concerns were partially addressed through the rebuttal and additional experiments. The authors provided new comparisons with relevant baselines (RCO, MultiCritique-RL), extended evaluations to more challenging reasoning tasks (creative writing, AIME, GPQA), clarified the intended distinction between “stronger supervision” and oracle-based reward signals, and reported robustness analyses under noisy feedback. These additions help clarify the paper’s positioning and strengthen the empirical support for the proposed method. However, some concerns are not fully resolved. In particular, the method’s reliance on the actor’s refinement capability may limit its effectiveness in settings with weaker or differently structured actors, and the applicability to domains without reliable evaluators remains only partially explored. In addition, some presentation and exposition issues persist, including clarity of terminology and organization of experimental details, which could be further improved.

**Reviewer Scores:**

Reviewer ao7e explicitly indicated an intention to increase the score from 4 to at least 6 following the additional experiments and clarifications. Based on the rebuttal, the other reviewers (4UWL, WXJP, FEHf) might either maintain their current scores or increase them moderately, as some of their concerns regarding novelty, empirical evaluation, and robustness were at least partially addressed.

---

### Decision · Program_Chairs · 2026-01-26

Accept (Poster)